# Inducible gene deletion reveals essentiality of protein kinases and a septation initiation network in *Candida albicans*

Bernardo Ramírez-Zavala[1], Ines Krüger[1], Sonja Schwanfelder[1], Johannes Lackner[1], Thomas Krüger[2], Olaf Kniemeyer[2], Joachim Morschhäuser[1]*

1 Institute of Molecular Infection Biology, University of Würzburg, Würzburg, Germany, 2 Department of Molecular and Applied Microbiology, Leibniz Institute for Natural Product Research and Infection Biology (HKI), Jena, Germany

* joachim.morschhaeuser@uni-wuerzburg.de

## Abstract

Protein kinases are key components of many signaling pathways that regulate cellular activities, and some of them are indispensable for the viability of cells. We used inducible gene deletion to assess the importance of a set of putative essential protein kinases for growth and viability of the pathogenic yeast *Candida albicans* and to get clues about the functions of uncharacterized essential kinases. We found that *bud32Δ*, *ctk1Δ*, *rio1Δ*, and *rio2Δ* mutants were viable but grew very slowly, explaining previous failures to generate homozygous deletion mutants. *PTK2* was essential, but under certain conditions *ptk2Δ* mutants remained viable and over time could acquire suppressor mutations in the Ptk2-dependent plasma membrane ATPase Pma1 that restored growth. Deletion of the uncharacterized *orf19.5376* was lethal and the null mutants formed pseudohyphae that lacked normal septa and eventually lysed, a phenotype that was phenocopied by auxin-induced protein depletion. The mutants were defective in septin organization, indicating that the *orf19.5376*-encoded kinase is functionally similar to the nonessential kinase Elm1 of *Saccharomyces cerevisiae*, but is indispensable for viability in *C. albicans*. Mutants lacking *orf19.3456*, which does not have a homolog in *S. cerevisiae*, were also nonviable and grew as aseptate, sometimes multinucleate hyphae before cell death. Co-immunoprecipitation followed by liquid chromatography-mass spectrometry identified a protein, encoded by the uncharacterized *orf19.193*, as a candidate regulatory subunit of the *orf19.3456*-encoded kinase, as mutants lacking this protein exhibited the same terminal phenotype as *orf19.3456* mutants. These results provide strong evidence that instead of using a mitotic exit network (MEN) with only two kinases (Cdc15 and Dbf2), as was previously thought, *C. albicans* regulates septum formation and cytokinesis via a septation initiation network (SIN), known from fission yeast and filamentous fungi, which contains a protein kinase cascade consisting of the upstream kinase Cdc15, the *orf19.3456*-encoded kinase, and the downstream kinase Dbf2.

**Data availability statement:** The mass spectrometry proteomics data have been deposited to the ProteomeXchange Consortium via the PRIDE [53] partner repository with the dataset identifier PXD070087.

**Funding:** This study was funded by the German Research Foundation (DFG) through the TRR 124 FungiNet, "Pathogenic fungi and their human host: Networks of Interaction," DFG project number 210879364, projects C2 to JM and Z2 to OK. Publication of the work was supported by the Open Access Publication Fund of the University of Würzburg. The funders had no role in study design, data collection and analysis, decision to publish, or preparation of the manuscript.

**Competing interests:** The authors have declared that no competing interests exist.

## Author summary

Elucidating the function of essential genes in the biology of an organism is challenging, because mutants lacking an essential gene are inviable and cannot be recovered by standard methods. To investigate the importance of a set of putative essential protein kinases for growth and viability of the pathogenic yeast *Candida albicans*, we generated mutants by forced, inducible gene deletion, which provides definite proof of whether a gene is essential or not. Some of the mutants turned out to be viable but grew very slowly, explaining previous failures to obtain homozygous deletion mutants. Other kinases were truly essential for viability, and the terminal phenotypes of the mutants before cell death provided insights into their function. Mutants lacking a previously uncharacterized kinase that has no homolog in the model yeast *Saccharomyces cerevisiae* were unable to grow as budding yeast cells and formed hyphae without septa that eventually lysed. Our results reveal that *C. albicans* and other pathogenic *Candida* species unexpectedly use a protein kinase signaling pathway that is known from fission yeast and filamentous fungi to regulate septum formation and cytokinesis during the cell cycle.

## Introduction

Protein kinases are key components of many signaling pathways that regulate basic cellular activities and the responses of cells to external signals. Deciphering the functions of individual protein kinases is therefore important to understand the regulatory networks that control the behavior of organisms and how they adapt to changes in their environment. The pathogenic yeast *Candida albicans* possesses 108 genes encoding known or predicted protein kinase catalytic subunits [1]. To enable systematic investigations of their roles in the biology and pathogenicity of this fungus, we previously had generated a comprehensive protein kinase deletion mutant library of the wild-type reference strain SC5314 [2,3]. We could successfully construct homozygous null mutants for 86 of these genes by sequentially deleting both alleles using the *SAT1*-flipping strategy [4,5]. For the remaining 22 protein kinase genes, only heterozygous mutants were obtained, possibly because they are essential for viability. Alternatively, the selection conditions might have prevented the recovery of homozygous mutants. Indeed, subsequent investigations showed that for two of these kinases, Snf1 and Ypk1, which were thought to be essential [6–10], null mutants could be generated by forced, inducible gene deletion [11,12]. The *snf1Δ* and *ypk1Δ* mutants exhibited a slow-growth phenotype that precluded their isolation under routine selection conditions. In the case of the *snf1Δ* mutants, growth was strongly improved at 37°C compared with the standard incubation temperature of 30°C [11].

The strategy of inducible gene deletion in *C. albicans*, which was originally established in our lab for an auxotrophic laboratory strain [13] and recently modified for use in wild-type strains [11], is outlined in Fig 1. A gene cassette containing a

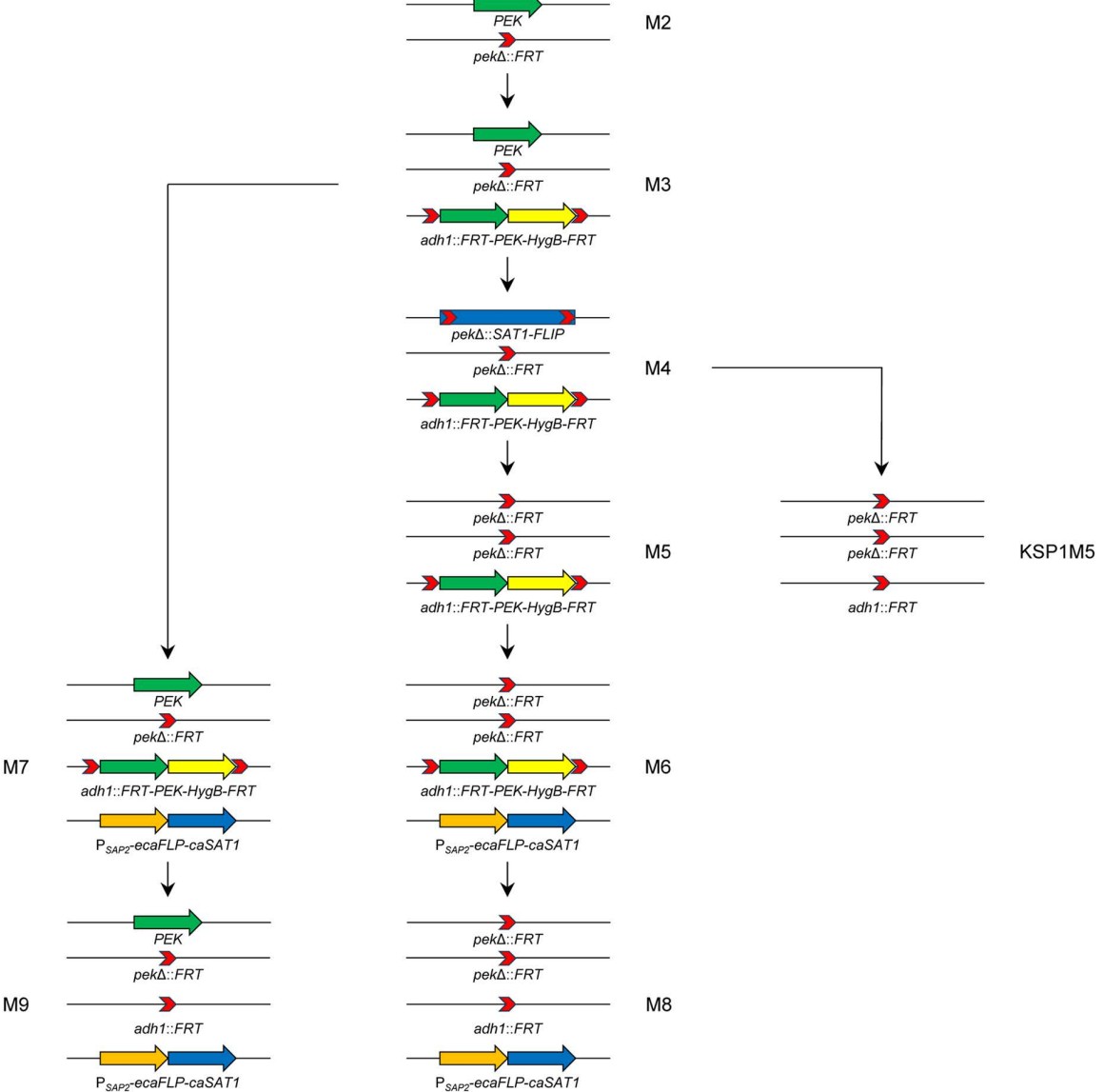

**Fig 1. Schematic illustrating the generation of the inducible gene deletion mutants.** Heterozygous mutants (M2) in which one allele of the putative essential kinase gene (*PEK*, green arrow) is already deleted were previously constructed. Ectopic integration of a functional *PEK* copy and the *HygB* selection marker (yellow arrow), flanked by FLP recognition sites (*FRT*, red chevrons), at the *ADH1* locus results in M3 mutants. Replacement of the second endogenous *PEK* allele by the *SAT1* flipper cassette (*SAT1-FLIP*, blue bar) and subsequent excision of the *SAT1* flipper cassette yields M4 and M5 mutants, respectively. A P~SAP2~-*ecaFLP* fusion (orange arrow) is then inserted into the *SAP2* locus with the help of the *caSAT1* selection marker (blue arrow) to obtain the conditional M6 mutants. The P~SAP2~-*ecaFLP* fusion is also integrated into the M3 mutants to produce control strains (M7) retaining one of the endogenous *PEK* alleles. If the *PEK* gene is not essential, forced deletion of the ectopically integrated copy after passage of the conditional mutants in *SAP2*-inducing YCB-BSA-YE medium yields viable *pekΔ* null mutants (M8) and corresponding control strains (M9). In the case of *KSP1*, excision of the *SAT1* flipper cassette from M4 mutants also produced derivatives that had simultaneously excised the ectopically integrated *KSP1* copy, which were kept as null mutants (KSP1M5).

functional copy of the putative essential gene and a hygromycin resistance marker (*HygB*), flanked by target sequences of the site-specific recombinase FLP, is inserted at an ectopic site in the genome of heterozygous mutants. The second endogenous allele of the target gene can then be deleted using the recyclable *SAT1* flipper cassette. Finally, the *ecaFLP*

(enhanced _Candida_-adapted _FLP_) gene, encoding a mutated version of the FLP recombinase with enhanced activity, is integrated under the control of the tightly regulated and efficiently inducible _SAP2_ promoter, using a nourseothricin resistance marker (_caSAT1_), to generate the desired conditional mutants. Passage in _SAP2_-inducing medium results in the excision of the FLP-deletable gene copy in the vast majority of the cells to produce an almost pure population of null mutants. Plating of these cells and incubation under any desired conditions provides definite proof of whether the gene is essential (no colony formation) or not. Furthermore, the phenotype and behavior of the null mutants before cell death may offer clues about the essential biological functions of the encoded protein. In the present study, we used this approach to assess the essentiality of a select set of protein kinases (as explained below) and obtain insights into the function of kinases that are indispensable for the viability of _C. albicans_ wild-type cells.

## Results

### Inducible deletion of putative essential protein kinase genes

We selected eight protein kinases (Table 1) for which we had not obtained homozygous deletion mutants during the construction of our library [2,3] to firmly establish by inducible gene deletion whether they are essential or not for viability in the wild-type reference strain SC5314. In a previous genome-wide _in vivo_ transposon mutagenesis study with a haploid _C. albicans_ strain, seven of these were deemed essential and only one of them (_KSP1_) was found to be dispensable [9]. Mutants in which _KSP1_ was inactivated by gene-specific transposon insertions had also been obtained before [6], but no homozygous _ksp1Δ_ mutants were recovered in another targeted approach [14]. Both of the latter two studies found _BUD32_ to be nonessential, but conflicting results were obtained for other kinases. _CTK1_ and _RIO1_ were considered as putative essential genes because no homozygous mutants were obtained by Blankenship _et al_. [6], but _ctk1Δ_ and _rio1Δ_ mutants were recently generated by Kramara _et al_. [14]. Conversely, _ptk2_ and _rio2_ transposon insertion mutants were obtained by Blankenship _et al_. [6] while no homozygous _ptk2Δ_ and _rio2Δ_ mutants could be generated by Kramara _et al_. [14]. For the two predicted protein kinases encoded by _orf19.3456_ and _orf19.5376_ no null mutants were obtained in all three studies. Interestingly, analysis of the GRACE collection of repressible mutants found none of the eight kinases to be essential, and only one of them (the _BUD32_ knock-down) exhibited a detectable growth defect [15] (see Table 1 for a summary).

**Table 1. Protein kinases analyzed in this study and reported putative essentiality[1].**

| Gene name | orf19 number | Tn mutagenesis [2] | UAU1 method [3] | Deletion [4] | Repression [5] |
|---|---|---|---|---|---|
| _BUD32_ | orf19.4252 | ESS | NE | NE | NE |
| _CTK1_ | orf19.1619 | ESS | ESS | NE | NE |
| _KSP1_ | orf19.4432 | NE | NE | ESS | NE |
| _PTK2_ | orf19.3415 | ESS | NE | ESS | NE |
| _RIO1_ | orf19.2320 | ESS | ESS | NE | NE |
| _RIO2_ | orf19.6369 | ESS | NE | ESS | NE |
| | orf19.3456 | ESS | ESS | ESS | NE |
| | orf19.5376 | ESS | ESS | ESS | NE |

[1]ESS, essential; NE, not essential

[2]ESS: No transposon insertions in haploid strain [9]

[3]ESS: No homozygous insertion mutants obtained by UAU1 method [6]

[4]ESS: No homozygous deletion mutants obtained [14]

[5]NE: Conditional mutants viable under repressive conditions [15]

For these eight putative essential kinases (PEKs), we generated inducible deletion mutants containing a single FLP-deletable gene copy (M6 mutants) as well as control strains that additionally retained one of the endogenous alleles (M7 mutants), as illustrated in Fig 1. All conditional mutants grew as well as the wild-type strain SC5314 and control strains, demonstrating that each of the single ectopically integrated gene copies was sufficient for normal growth under standard conditions (S1 Fig). After the induced gene deletion, five of the eight null mutants (bud32Δ, ctk1Δ, ksp1Δ, rio1Δ, rio2Δ) remained viable and produced colonies upon subsequent plating and incubation on YPD medium (Fig 2). In the case of KSP1, this became evident already after the excision of the SAT1 flipper cassette from the second endogenous allele, as some of the analyzed descendants had simultaneously excised the ectopically integrated KSP1 copy and become null mutants (see Fig 1). The ksp1Δ mutants grew as well as the wild-type strain SC5314 and control strains retaining the ectopically integrated KSP1 copy (Figs 2, S2; the plates shown in Fig 2 contain both types of M4 derivatives), demonstrating that our previous failure to obtain homozygous ksp1Δ mutants had technical reasons. The bud32Δ, rio1Δ, and especially the ctk1Δ and rio2Δ mutants generated by forced deletion of the ectopically integrated last gene copy grew poorly on rich medium, with improved growth at 37°C compared to 30°C, explaining why homozygous mutants were not recovered in our previous efforts (Figs 2, S2).

In contrast, after deletion of the last copy of PTK2, orf19.3456, and orf19.5376, the null mutants were unable to grow under standard conditions (Fig 3). The induced gene deletion occurred with high efficiency, since <0.1% of the cells grown in the inducing medium produced normal colonies after plating on rich medium, in contrast to the control strains in which the majority of cells yielded colonies (Table 2). However, upon prolonged incubation, the ptk2Δ mutants started to form visible colonies of different sizes, especially at 37°C in a $CO_2$ incubator, suggesting that the mutants survived the gene deletion and over time accumulated suppressor mutations that enabled growth (Fig 3). When restreaked on fresh medium, cells from large colonies grew well also at 30°C (S3A Fig), indicating that incubation in a $CO_2$ incubator only facilitated the acquisition of suppressor mutations. We confirmed that the ectopically integrated PTK2 copy had been correctly excised by FLP-mediated recombination in four well-growing ptk2Δ mutants, two each derived from conditional mutants M6A and M6B (S3 Fig). Ptk2 has two largely redundant orthologs in the model yeast Saccharomyces cerevisiae, Ptk1 and Ptk2, which are required for the activity of the essential plasma membrane $H^+$-ATPase Pma1 by phosphorylating its autoinhibitory C-terminal domain at S911 and T912 [16]. The corresponding residues S888 and T889 are conserved in C. albicans Pma1 and have been found to be phosphorylated in two phosphoproteome studies [17,18]. Deletion of the autoinhibitory domain, and also a G648S substitution in Pma1 that interferes with the self-inhibition, bypass the requirement for Ptk1/2 in S. cerevisiae and render ptk1Δ ptk2Δ double mutants viable [16]. We therefore sequenced the PMA1 alleles of the four ptk2Δ suppressor mutants shown in S3 Fig. Intriguingly, three of the four clones contained mutations in PMA1. Clone A1 was heterozygous for a T1760C mutation resulting in a G587A substitution in Pma1, A2 was homozygous for a C1709T mutation resulting in an A570V substitution, and B2 was heterozygous for a T2647C mutation resulting in a S883P substitution. Clone B1 did not contain a mutation in the PMA1 coding sequence and retained the wild-type alleles. These results indicate that the essential function of Ptk1/2 in Pma1 activation in S. cerevisiae is conserved in C. albicans Ptk2, and suppressor mutations in Pma1, but also other mutations, enable growth of ptk2Δ mutants. Some slow-growing colonies also appeared after prolonged growth of orf19.3456Δ and orf19.5376Δ mutants, but at a much lower frequency (Fig 3, which also shows a rare normally growing colony from a cell that had escaped FLP-mediated orf19.5376 deletion), indicating that the encoded kinases are essential for viability of the wild-type strain SC5314.

## Phenotype of orf19.3456Δ and orf19.5376Δ mutants

We focused our further efforts on the two uncharacterized essential protein kinases encoded by orf19.3456 and orf19.5376. We first monitored the terminal phenotype of the null mutants by inoculating them onto YPD agar after the induced gene deletion and observing the cells by time-lapse video microscopy. In contrast to the wild-type strain SC5314 and the M7 control strains, which grew as budding yeasts under these conditions, both mutants were unable to maintain the normal yeast morphology after 2–3 initial cell doublings that were likely enabled by remaining gene product (S1-S5

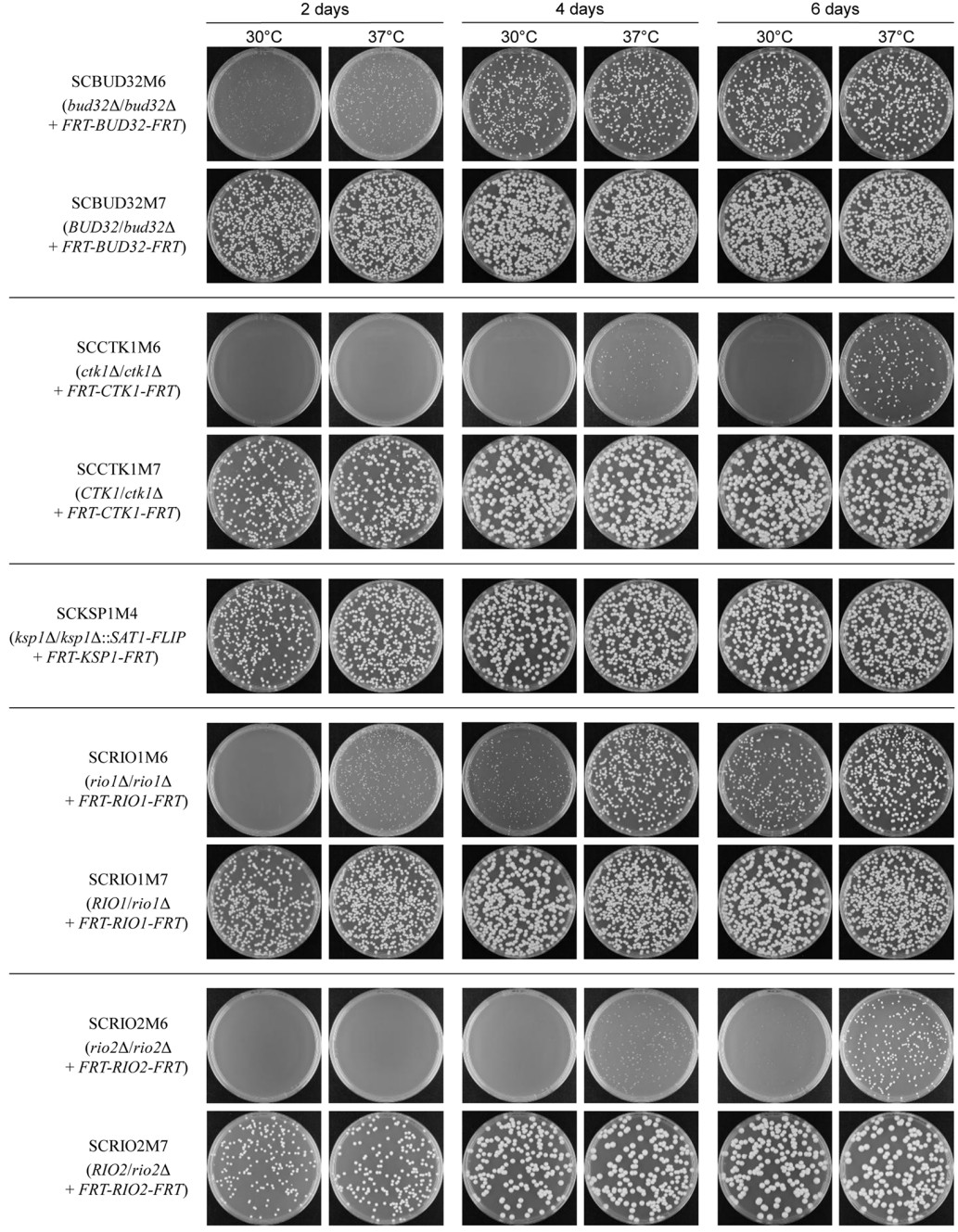

**Fig 2. Growth of viable *pek*Δ null mutants.** The conditional M6 mutants and M7 control strains were passaged overnight in YCB-BSA-YE medium to induce FLP-mediated excison of the ectopically integrated *PEK* copy. For *KSP1* the M4 mutants were used. Appropriate dilutions of the cultures were plated on YPD medium and incubated at 30°C and at 37°C. Photographs were taken after 2, 4, and 6 days. The two independently generated series of strains behaved identically and only one of them is shown in each case.

Videos, Figs 4, S4). The *orf19.3456*Δ mutants first formed aberrantly shaped cells and then often grew as thin hyphae; over time the filaments stopped elongating and the cells eventually lysed. The *orf19.5376*Δ mutants produced pseudohyphae with constrictions between highly elongated cells, ultimately followed by cell lysis.

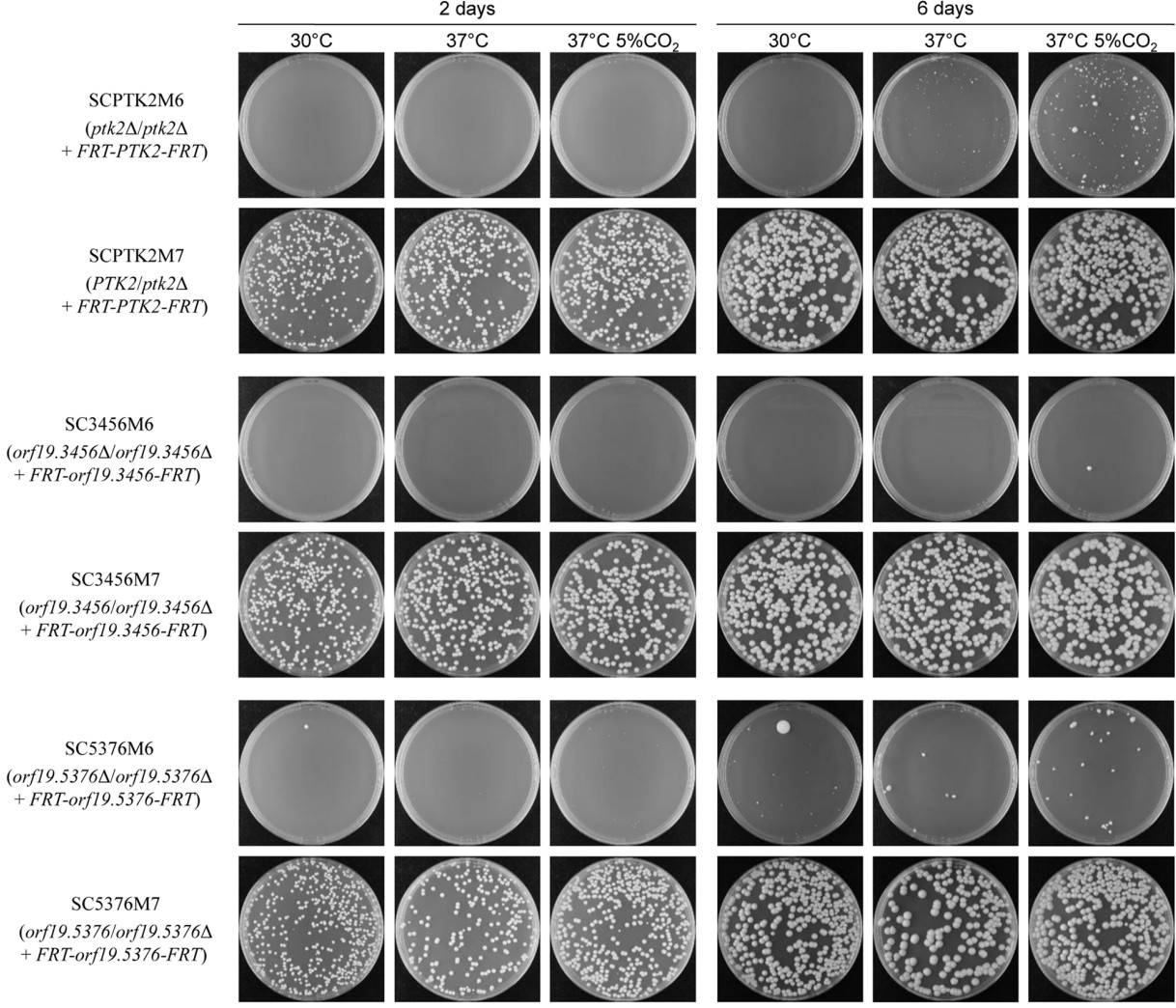

**Fig 3. *PTK2*, *orf19.3456*, and *orf19.5376* are essential.** The conditional M6 mutants and M7 control strains were passaged overnight in YCB-BSA-YE medium to induce FLP-mediated excison of the ectopically integrated *PEK* copy. Appropriate dilutions of the cultures were plated on YPD medium and incubated at 30°C, 37°C, and in a $CO_2$ incubator at 37°C. Photographs were taken after 2 and 6 days. The two independently generated series of strains behaved identically and only one of them is shown in each case.

To further characterize the filamentous cells of the mutants, the strains were grown in YPD liquid medium after the induced gene deletion and then fixed and treated with calcofluor white, to stain chitin in cell walls and septa, or with DAPI, to visualize nuclei (Fig 5). In the *orf19.3456Δ* mutants, septa were only detectable between yeast cells that remained attached to one another, but no septa were visible in the long filaments (Fig 5A). While aberrantly shaped yeast cells of the *orf19.3456Δ* mutants often contained multiple nuclei, nuclear staining became diffuse and extended in the long filaments, indicating that the mutants were defective in nuclear segregation and distribution (Fig 5B). The pseudohyphae of the *orf19.5376Δ* mutants showed no or mislocalized chitin at the constrictions between cells, pointing to a defect in septum formation (Fig 5A) that resulted in some cells having no or multiple nuclei (Fig 5B, compare with serum-induced wild-type hyphae in S5 Fig). The M7 control strains that retained a wild-type allele after excision of the FLP-deletable

**Table 2. Efficiency of induced gene deletion.**

| Strain [1] | Description | Total cells/ml | CFU/ml | CFU/total cells |
|---|---|---|---|---|
| SCPTK2M6A (1) | Conditional *ptk2Δ* mutant | $1.3 \times 10^9$ | $2.5 \times 10^5$ | $1.9 \times 10^{-4}$ |
| SCPTK2M6A (2) | Conditional *ptk2Δ* mutant | $1.2 \times 10^9$ | $1.6 \times 10^5$ | $1.3 \times 10^{-4}$ |
| SCPTK2M6B (1) | Conditional *ptk2Δ* mutant | $1.4 \times 10^9$ | $2.6 \times 10^5$ | $1.9 \times 10^{-4}$ |
| SCPTK2M6B (2) | Conditional *ptk2Δ* mutant | $1.2 \times 10^9$ | $2.8 \times 10^5$ | $2.4 \times 10^{-4}$ |
| SCPTK2M7A (1) | Control strain | $1.2 \times 10^9$ | $8.2 \times 10^8$ | 0.67 |
| SCPTK2M7A (2) | Control strain | $1.3 \times 10^9$ | $7.8 \times 10^8$ | 0.62 |
| SCPTK2M7B (1) | Control strain | $1.1 \times 10^9$ | $7.7 \times 10^8$ | 0.72 |
| SCPTK2M7B (2) | Control strain | $1.2 \times 10^9$ | $8.3 \times 10^8$ | 0.70 |
| | | | | |
| SC3456M6A (1) | Conditional *orf19.3456Δ* mutant | $1.4 \times 10^9$ | $3.5 \times 10^5$ | $1.9 \times 10^{-4}$ |
| SC3456M6A (2) | Conditional *orf19.3456Δ* mutant | $1.4 \times 10^9$ | $2.3 \times 10^5$ | $1.3 \times 10^{-4}$ |
| SC3456M6B (1) | Conditional *orf19.3456Δ* mutant | $1.2 \times 10^9$ | $1.4 \times 10^5$ | $1.9 \times 10^{-4}$ |
| SC3456M6B (2) | Conditional *orf19.3456Δ* mutant | $1.3 \times 10^9$ | $2.7 \times 10^5$ | $2.4 \times 10^{-4}$ |
| SC3456M7A (1) | Control strain | $9.4 \times 10^8$ | $9.4 \times 10^8$ | 0.67 |
| SC3456M7A (2) | Control strain | $1.3 \times 10^9$ | $9.1 \times 10^8$ | 0.62 |
| SC3456M7B (1) | Control strain | $1.2 \times 10^9$ | $1.1 \times 10^9$ | 0.72 |
| SC3456M7B (2) | Control strain | $1.2 \times 10^9$ | $9.2 \times 10^8$ | 0.70 |
| | | | | |
| SC5376M6A (1) | Conditional *orf19.5376Δ* mutant | $1.3 \times 10^9$ | $7.3 \times 10^5$ | $5.5 \times 10^{-4}$ |
| SC5376M6A (2) | Conditional *orf19.5376Δ* mutant | $1.6 \times 10^9$ | $4.6 \times 10^5$ | $3.0 \times 10^{-4}$ |
| SC5376M6B (1) | Conditional *orf19.53576Δ* mutant | $1.2 \times 10^9$ | $6.0 \times 10^5$ | $5.1 \times 10^{-4}$ |
| SC5376M6B (2) | Conditional *orf19.5376Δ* mutant | $1.4 \times 10^9$ | $8.0 \times 10^5$ | $5.8 \times 10^{-4}$ |
| SC5376M7A (1) | Control strain | $1.2 \times 10^9$ | $1.1 \times 10^9$ | 0.91 |
| SC5376M7A (2) | Control strain | $1.3 \times 10^9$ | $7.7 \times 10^8$ | 0.58 |
| SC5376M7B (1) | Control strain | $1.1 \times 10^9$ | $8.5 \times 10^8$ | 0.75 |
| SC5376M7B (2) | Control strain | $1.3 \times 10^9$ | $1.0 \times 10^9$ | 0.81 |

[1]Two independent colonies of each strain were tested.

gene copy behaved like the parental strain SC5314 (S6 Fig). Altogether, these results indicate that both *orf19.3456Δ* and *orf19.5376Δ* mutants have severe cytokinesis defects that ultimately result in cell death.

### The kinase activity of the proteins encoded by *orf19.3456* and *orf19.5376* is essential for viability

To investigate if the essential function of the proteins encoded by *orf19.3456* and *orf19.5376* depends on their predicted kinase activity, we generated strains that after loss of the FLP-deletable wild-type allele retained a mutated, "kinase-dead" allele at the endogenous locus (see materials and methods and S1 Table). These strains behaved like the conditional null mutants; they could not generate colonies after the deletion of the wild-type allele (S7A Fig) and had the same terminal phenotype as the null mutants (S6-S7 Videos and S7B Fig). We confirmed that the mutated proteins were produced by introducing 3xHA-tagged wild-type and kinase-dead alleles at the endogenous locus into strains with an additional wild-type allele (to ensure viability). Western blotting showed that wild-type and kinase-dead proteins were produced at comparable levels (S7C Fig). Furthermore, HA-tagging did not detectably compromise protein function, since replacement of the remaining wild-type allele in the heterozygous M2 mutants by a 3xHA-tagged allele did not affect growth (S7D Fig). These results provide evidence that the kinase activity of the predicted protein kinases encoded by *orf19.3456* and *orf19.5376* is essential for viability in the *C. albicans* wild-type strain SC5314.

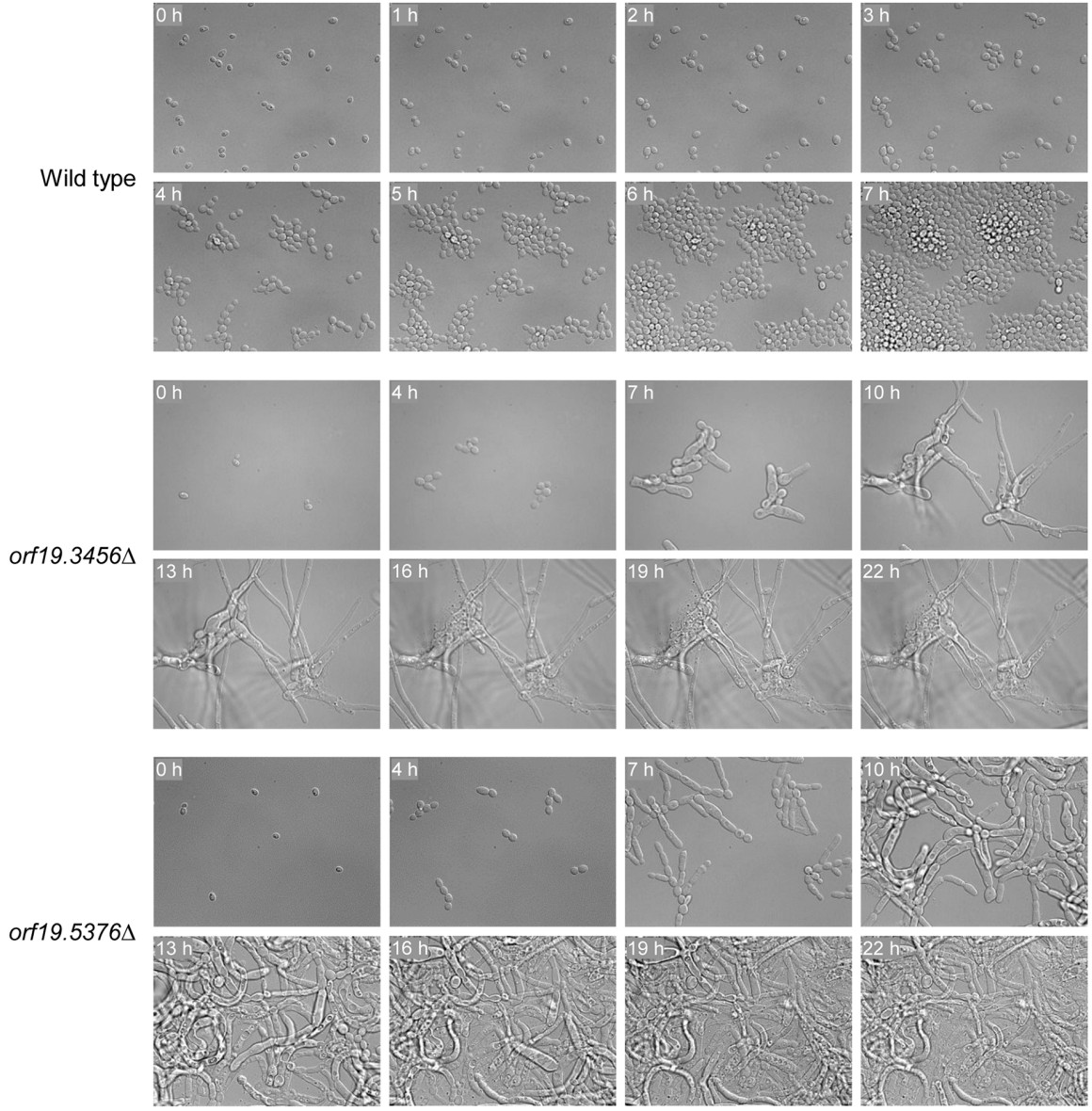

**Fig 4. Terminal phenotype of *orf19.3456Δ* and *orf19.5376Δ* mutants.** The conditional M6 mutants were passaged overnight in YCB-BSA-YE medium to induce FLP-mediated excison of the ectopically integrated gene copy. The cultures were diluted in water, transferred to a 35 mm culture dish, covered with YPD agar, and incubated at 30°C. Images were taken every 5 min with a DMI6000 Leica inverted microscope (S1-S3 Videos). The figure shows photographs of the cells at the indicated time points. The wild-type strain SC5314 was treated identically and is included for comparison. M7 control strains are shown in S4-S5 Videos and S6 Fig.

### *orf19.5376* encodes a functional homolog of *Saccharomyces cerevisiae* Elm1 that regulates septin localization and cytokinesis

The protein kinase encoded by *orf19.5376* is the closest *C. albicans* homolog of *S. cerevisiae* Elm1, which has a redundant role with two other kinases, Sak1 and its paralog Tos3, in the activation of Snf1, a protein kinase that is required for adaptation to glucose limitation and utilization of alternative carbon sources [19–21]. In *C. albicans*, Snf1 phosphorylation in its activation loop is largely abolished in *sak1Δ* mutants, indicating that *orf19.5376* cannot compensate for the loss of

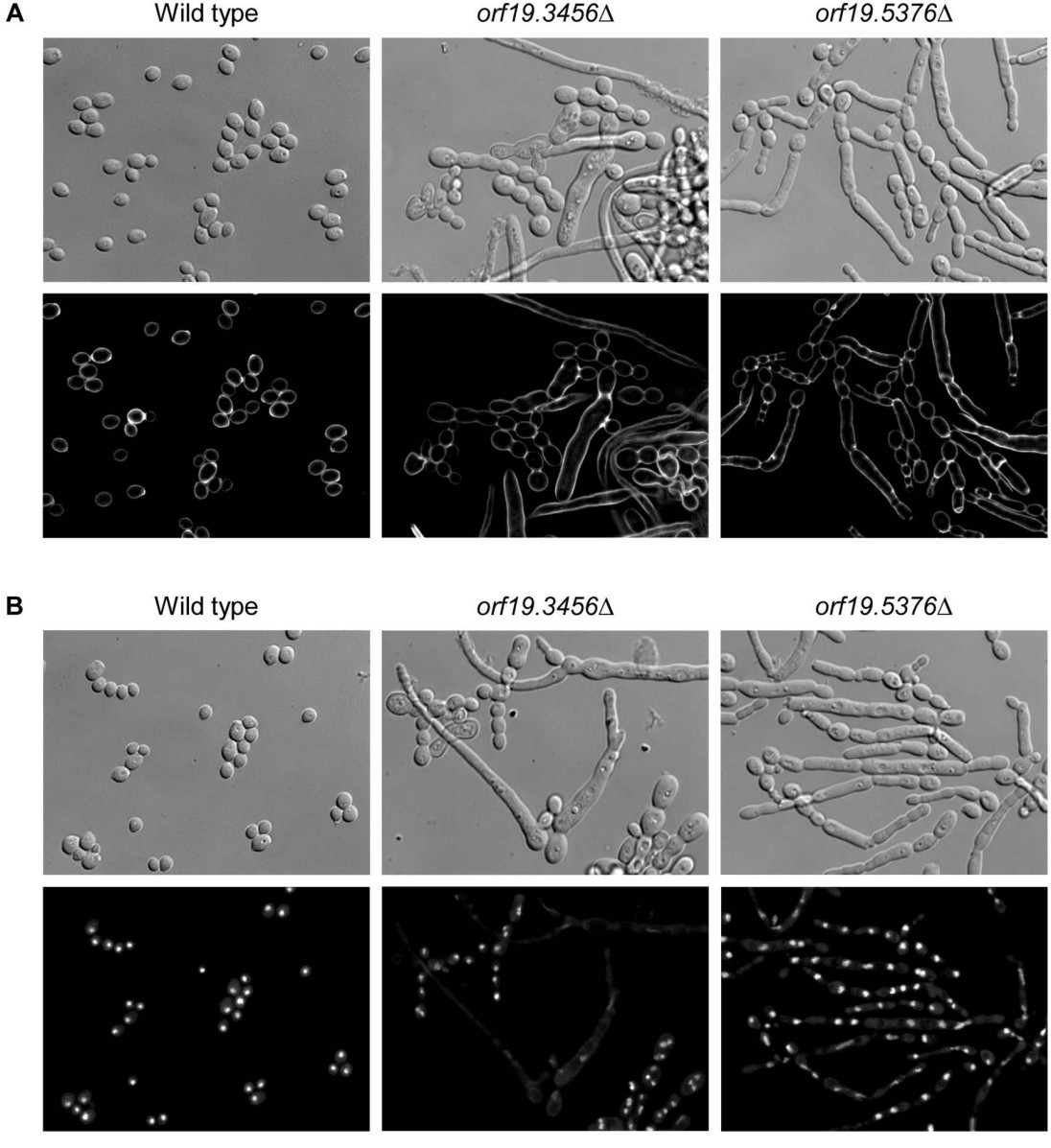

**Fig 5. *orf19.3456Δ* and *orf19.5376Δ* mutants have cytokinesis defects.** YCB-BSA-YE overnight cultures of the wild-type strain SC5314 and the conditional M6 mutants were diluted 1:100 in YPD medium and grown at 30°C. Aliquots of the cultures were taken after 12 h and fixed with formaldehyde. Cells were washed with PBS and stained with calcofluor white (A) or DAPI (B). Cells were imaged by DIC and fluorescence microscopy.

Sak1 and does not have a significant, if any, role in Snf1 activation [3]. However, Elm1 has additional functions in the yeast cell cycle. It is localized at the bud neck where it controls the assembly of septin filaments into the hourglass structure and their reorganization into the double ring structure at the division site; absence of Elm1 results in a mitotic delay [22–26]. *S. cerevisiae elm1Δ* mutants display a highly elongated cell morphology and, in some genetic backgrounds, constitutive pseudohyphal growth, because the cells remain attached to each other after cytokinesis [27]. These phenotypes resemble those of *C. albicans orf19.5376Δ* mutants (Fig 4 and S3 Video). To study *orf19.5376* function further, we tested

if the *orf19.5376Δ* mutant phenotype could also be brought about by auxin-inducible protein depletion [28], and possibly faster than by forced gene deletion. To this aim, the remaining wild-type *orf19.5376* allele in the two independently generated heterozygous M2 mutants was fused with the AID* cassette (see materials and methods). Log-phase cells of the resulting strains were treated with 1 μM of the auxin analog 5-Ad-IAA and samples were taken at different time points for detection of the tagged proteins by Western blotting. Fig 6A shows that the degron-tagged kinase, which can be detected with an anti-HA antibody, was successfully removed within 15 min of treatment. To determine the phenotypic consequences of protein depletion, cells grown in YPD liquid medium were transferred to YPD agar containing 1 μM 5-Ad-IAA and observed by video microscopy. As can be seen in S8 Video and Fig 6B, the cells exhibited the same pseudohyphal phenotype as the induced gene deletion mutants, only that it was induced more rapidly, presumably because of the more immediate effect of protein depletion (identically treated, untagged control cells are shown in S9 Video). Furthermore, staining of cells grown in YPD liquid medium showed that auxin-induced protein depletion resulted in the same defects in septum formation and nuclear distribution as seen in the gene deletion mutants (S8 Fig). This was further corroborated by monitoring septin localization in cells containing a *GFP*-tagged *CDC3* allele. In contrast to auxin-treated wild-type cells, which showed the typical septin ring between mother and daughter cells as well as the double ring in large-budded cells (S9 Fig), Cdc3 was largely absent at the restrictions between cells and relocalized to the pseudohyphal tip in the auxin-induced mutants (Fig 6C), similar to the mislocalization of septins in *S. cerevisiae elm1Δ* mutants [22,26,29,30]. For comparison, serum-induced hyphae of the wild-type strain showed normal septa, nuclei distribution, and septin rings (S10 Fig). Together, these observations demonstrate that the kinase encoded by *orf19.5376* controls septin localization and cytokinesis in *C. albicans*, indicating that it is indeed a functional ortholog of *S. cerevisiae* Elm1. However, while *S. cerevisiae elm1Δ* mutants are viable, the morphologial defects caused by the absence of this kinase are much more severe and eventually lethal in *C. albicans*.

**The protein kinase encoded by *orf19.3456* is part of a septation initiation network in *C. albicans***

The protein kinase encoded by *orf19.3456* does not have an ortholog in *S. cerevisiae*, but one of its homologs in the fission yeast *Schizosaccharomyces pombe* is the protein serine/threonine kinase Sid1, a component of the septation initiation network (SIN) that coordinates cytokinesis with chromosome segregation [31,32]. The SIN includes a kinase cascade, Cdc7-Sid1-Sid2, that is positioned at the spindle pole body (SPB) by the GTPase Spg1. The SIN becomes active at the new SPB (by the recruitment of Sid1) when the mitotic cyclin-dependent kinase (CDK) is inactivated in late mitosis. Under restrictive conditions, *sid1* as well as other *sin* mutants undergo nuclear division without septation and form elongated, multinucleate cells that eventually lyse [31]. A related signaling pathway in *S. cerevisiae* is the mitotic exit network (MEN), which promotes exit from mitosis by triggering the release of the phosphatase Cdc14 from the nucleolus to antagonize the mitotic CDKs [33,34]. The MEN contains homologs of many of the SIN components, including the Spg1 homolog Tem1, the Cdc7 homolog Cdc15, and the Sid2 homologs Dbf2/Dbf20, but it lacks a Sid1 homolog and Dbf2 is directly activated by Cdc15 [35]. *S. cerevisiae* mutants lacking MEN proteins arrest in late anaphase and fail to inactivate CDKs and initiate cytokinesis [34].

Sid1 has comparable homology to several *C. albicans* kinases, not only that encoded by *orf19.3456* (62.1% similarity in the kinase domain), e.g., Kic1 (64.1%), Cla4 (60.6%), Sps1 (58.5%), or Cdc15 (58.1%). However, the *orf19.3456Δ* mutant phenotype (multinucleate cells, hyphae without septa) suggested that the encoded kinase could be a Sid1 ortholog and that *C. albicans*, despite being a budding yeast like *S. cerevisiae*, possesses a signaling pathway that is more related to the SIN of *S. pombe* than to the MEN of *S. cerevisiae*. We tested if the *orf19.3456Δ* mutant phenotype could be reproduced by auxin-induced protein depletion and replaced the remaining wild-type *orf19.3456* allele in the heterozygous M2 mutants by an AID*-tagged copy. Fig 7A shows that the degron-tagged protein became undetectable within 15 min after the addition of 1 μM 5-Ad-IAA to log-phase cells. Similarly to the gene deletion mutants, the cells were unable to grow as budding yeasts and instead formed hyphae without constrictions, and this phenotype was rapidly induced upon protein

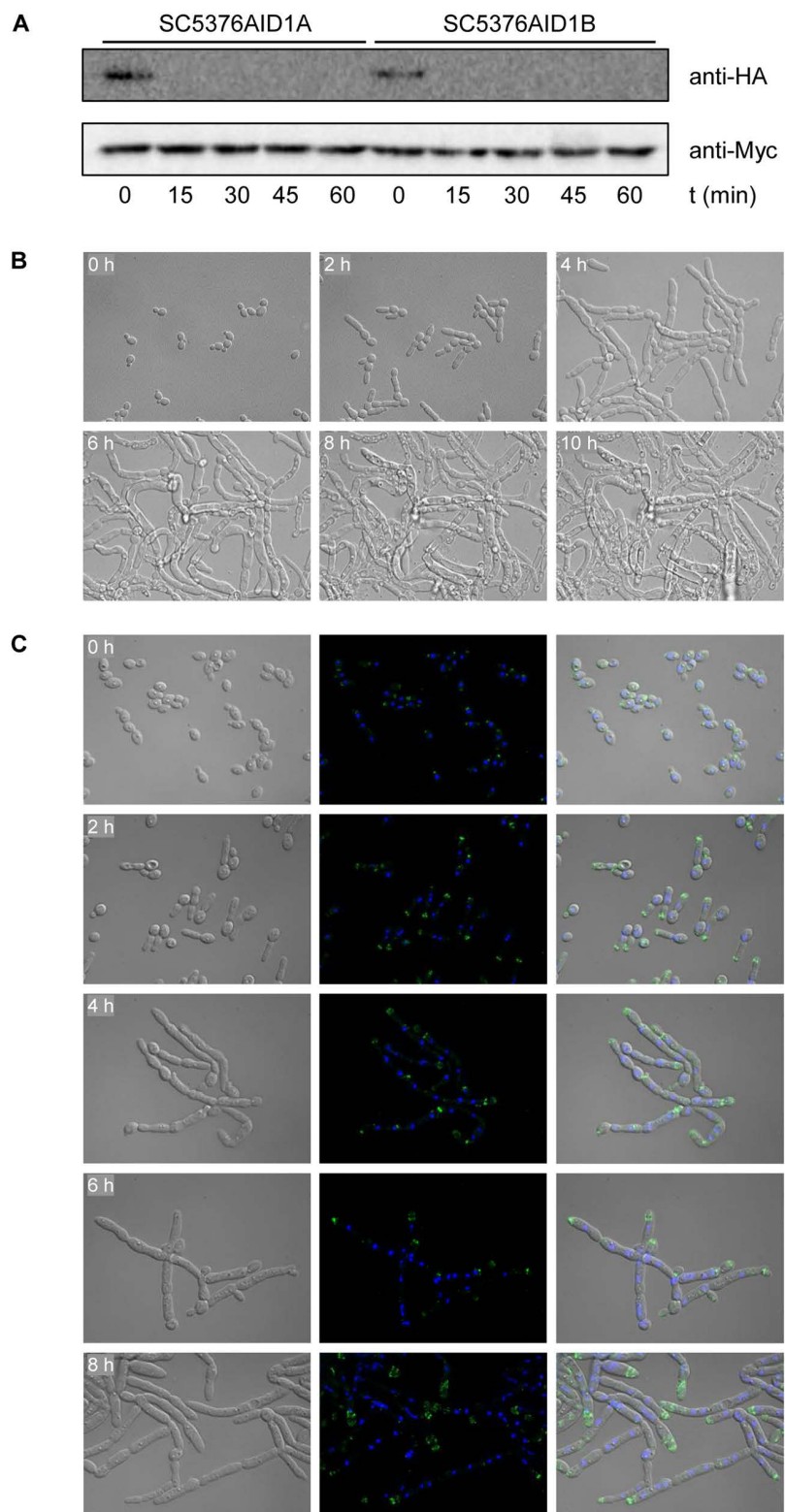

**Fig 6. Phenotypic consequences of auxin-induced degradation of the *orf19.5376*-encoded kinase. (A)** Strains containing a single *orf19.5376* copy fused to the AID* cassette were grown to log phase in YPD medium. Samples were taken before and at the indicated time points after addition of 1 µM

5-Ad-IAA and analyzed by Western blotting. The degron-tagged kinase was detected with an anti-HA antibody and the Tir protein with an anti-Myc antibody. **(B)** YPD-grown cells containing the degron-tagged kinase were transfered to a 35-mm culture dish, covered with YPD + 1 μM 5-Ad-IAA agar, and incubated at 30°C. Images were taken every 5 min for 24 h with a Leica DMI6000 microscope (S8 Video). The figure shows photographs of the cells at the indicated time points. **(C)** A YPD overnight culture of the auxin-inducible mutants containing a *GFP*-tagged *CDC3* allele was diluted 1:100 in YPD + 1 μM 5-Ad-IAA and grown at 30°C. Aliquots of the culture were taken every 2 h and fixed with paraformaldehyde. Cells were washed with PBS, stained with DAPI, and imaged by DIC (left panels) and fluorescence microscopy (middle panels). The figure shows photographs of the cells at the indicated time points, including overlays of the DIC and fluorescence micrographs (right panels).

depletion (S10 Video and Fig 7B). The video also shows that the hyphae originated from buds that had become larger than their mother cells (identically treated, untagged control cells are shown in S11 Video). Moreover, the defect in septum formation and a sometimes multinucleate nature of aseptate hyphae were evident in these mutants (S11 Fig). In cells containing a *GFP*-tagged *CDC3* allele, the septin became mislocalized soon after the auxin-induced protein depletion and was either absent in hyphae or found at apparently random sites without forming a ring structure Fig 7C).

Several components of the MEN/SIN pathway have already been studied in *C. albicans* and, in analogy to *S. cerevisiae*, described as parts of the MEN [36–39]. Indeed, the GTPase Tem1, the kinases Cdc15 and Dbf2, and the phosphatase Cdc14 were thought to constitute all key components of the MEN signaling cascade in *C. albicans* [36]. *TEM1*, *CDC15*, and *DBF2* are essential genes, and their functions have been investigated by expressing them under the control of repressible promoters. The *tem1* and *cdc15* knock-down mutants formed hyphae without septa [36,39], similar to our *orf19.3456* mutants. In contrast, *dbf2* knock-down mutants showed a different phenotype. They were unable to make septa, but instead of forming hyphae produced chains of cells that failed to separate after promoter shut-off [38]. Cells expressing an HA-tagged *DBF2*, whose function was partially compromised, exhibited an even stronger phenotype under repressive conditions. They arrested as pairs of large budded cells without a septum after DNA replication and displayed a nuclear segregation defect due to improper mitotic spindle organization [38]. To investigate whether the *orf19.3456*-encoded kinase and the putative downstream kinase Dbf2 indeed have different functions in the *C. albicans* cell cycle or the different mutant phenotypes were caused by the experimental approaches (gene repression versus gene deletion/ protein depletion), we generated auxin-inducible *dbf2* mutants of the wild-type strain SC5314. Dbf2-depleted cells exhibited the same phenotype as the *orf19.3456* mutants; they formed aseptate hyphae that often contained two elongated nuclei, suggesting a nuclear segregation defect after they had entered a new cell cycle (Fig 8 and S12 Video; auxin-treated, untagged control cells are shown in S13 Video and S12 Fig). These results suggest that Tem1, Cdc15, the *orf193456*-encoded kinase, and Dbf2 act in the same signaling pathway and their loss results in the same terminal phenotype.

### *orf19.193* encodes a putative regulatory subunit of the *orf19.3456*-encoded kinase

Sid1 of *S. pombe* is bound to a regulatory subunit that is required for its activity [31]. To identify potential interaction partners of the *orf19.3456*-encoded kinase, we performed co-immunoprecipitation (Co-IP) experiments with derivatives of strain SC5314 in which both *orf19.3456* alleles were HA-tagged. Liquid chromatography-mass spectrometry identified 105 proteins (including the tagged kinase itself) that were significantly enriched in immunoprecipitates of the tagged strains compared to the untagged wild-type control strain (S2 Table; extended dataset provided in S3 Table). Among those, one protein stood out because of its > 1,000-fold increased abundance. It is encoded by the putative essential *orf19.193*, the function of which is unknown. However, according to the *Candida* Genome Database, the protein belongs to the same family as Cdc14, the regulatory subunit of Sid1 in *S. pombe* (not to be confounded with the phosphatase Cdc14 of *S. cerevisiae* and *C. albicans*), and a subsequent BLAST search of the *C. albicans* genome with *S. pombe* Cdc14 identified *orf19.193* as the best hit. We confirmed the binding of the *orf19.193*-encoded protein to the *orf19.3456*-encoded kinase in Co-IP experiments with strains containing Myc-tagged *orf19.3456* and/or HA-tagged *orf19.193* (see S4 Table). Fig 9 shows that the *orf19.193*-encoded protein was specifically immunoprecipitated by the Myc-tagged kinase (lanes 5–8), but not in strains with the untagged kinase (lanes 1 and 2).

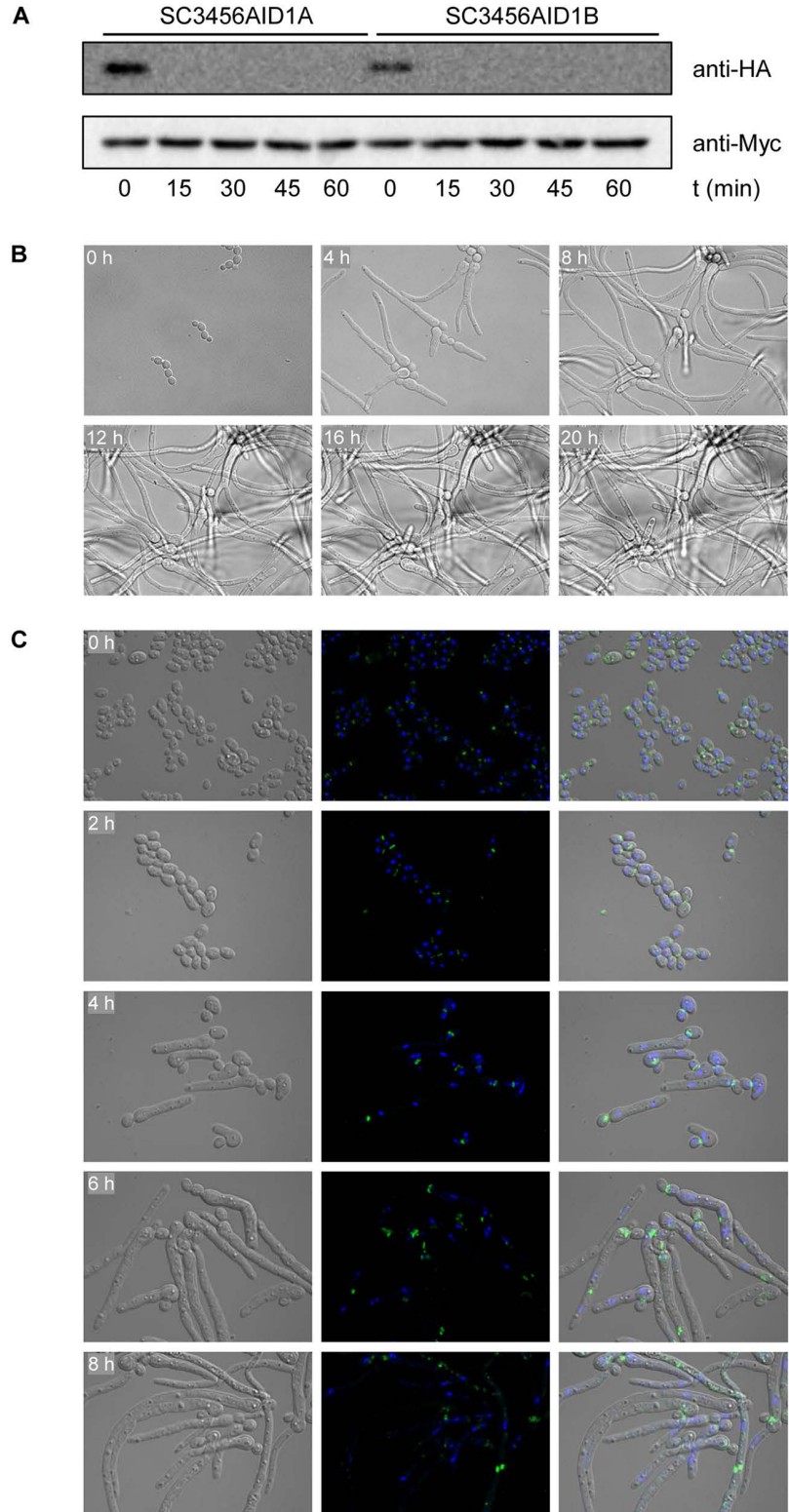

**Fig 7. Phenotypic consequences of auxin-induced degradation of the *orf19.3456*-encoded kinase. (A)** Strains containing a single *orf19.3456* copy fused to the AID* cassette were grown to log phase in YPD medium. Samples were taken before and at the indicated time points after addition of 1 µM 5-Ad-IAA and analyzed by Western blotting with anti-HA and anti-Myc antibodies. **(B)** YPD-grown cells containing the degron-tagged kinase were

transfered to a 35-mm culture dish, covered with YPD + 1 µM 5-Ad-IAA agar, and incubated at 30°C. Images were taken every 5 min for 24 h with a Leica DMI6000 microscope (S10 Video). The figure shows photographs of the cells at the indicated time points. **(C)** A YPD overnight culture of the auxin-inducible mutants containing a *GFP*-tagged *CDC3* allele was diluted 1:100 in YPD + 1 µM 5-Ad-IAA and grown at 30°C. Aliquots of the culture were taken every 2 h and fixed with paraformaldehyde. Cells were washed with PBS, stained with DAPI, and imaged by DIC (left panels) and fluorescence micros-copy (middle panels). The figure shows photographs of the cells at the indicated time points, including overlays of the DIC and fluorescence micrographs (right panels). Note that the hyphae were deformed after the fixation and centrifugation steps (compare with the live hyphae in **(B)**).

We reasoned that, if *orf19.193* encodes a regulatory subunit that is essential for the activity of the *orf19.3456*-encoded kinase, *orf19.193* mutants should exhibit the same phenotype as *orf19.3456* mutants. Either of the two *orf19.193* alleles could be deleted in the wild-type strain SC5314, but no homozygous mutants were obtained among 24 second-round transformants of the two types of heterozygous mutants, providing further evidence that *orf19.193* is an essential gene. We therefore generated auxin-inducible conditional mutants in which the remaining wild-type *orf19.193* allele was fused with the AID* cassette. Strikingly, following depletion of the degron-tagged protein, the cells grew as aseptate hyphae containing elongated nuclear masses, just like the *orf19.3456* mutants (Fig 10 and S14 Video; auxin-treated, untagged control cells are shown in S15 Video and S13 Fig). This phenotype provided additional evidence that *C. albicans* regulates cytokinesis *via* a SIN pathway that includes the *orf19.3456*-encoded kinase and its regulatory subunit in addition to the upstream kinase Cdc15 and the downstream kinase Dbf2.

## Discussion

The results presented in this work illustrate that inducible gene deletion is a powerful method to determine gene essen-tiality in *C. albicans*. We found that the protein kinases Bud32, Ctk1, Rio1, and Rio2 are not essential for viability of the wild-type strain SC5314, but *bud32Δ*, *ctk1Δ*, *rio1Δ*, and *rio2Δ* mutants grew very slowly, explaining why homozygous deletion mutants for these genes were not recovered in our previous efforts [2] and why no insertions in these genes were found in a genome-wide *in vivo* transposon mutagenesis study with a haploid *C. albicans* strain [9]. A particular advantage of the induced gene deletion is that it occurs in almost every individual cell of the conditional mutants, resulting in a pop-ulation of independent null mutants whose growth can be compared by plating for single colonies immediately after the gene deletion. The small size of all colonies generated by individual *bud32Δ*, *ctk1Δ*, *rio1Δ*, and *rio2Δ* cells after prolonged incubation demonstrated that absence of the corresponding kinases results in poor growth even under optimal conditions (see Fig 2). In this respect, we note that Kramara et al. [14] did not observe a fitness defect of *ctk1Δ* and *rio1Δ* mutants generated in strain SN250, an auxotrophic derivative of SC5314, whereas our induced *ctk1Δ* and *rio1Δ* deletion mutants exhibited severely reduced growth (Figs 2, S2), indicating that a suppressor mutation may have allowed normal growth of the former mutants. The occurrence of suppressor mutations that enable (normal) growth of otherwise nonviable or poorly growing mutants is indeed an issue that must be kept in mind, especially when rare homozygous mutants are obtained using traditional gene deletion methods. It also became evident when we recovered *ptk2Δ* mutants after the induced gene deletion. Colonies of highly variable size appeared after prolonged incubation at 37°C, indicating that the *ptk2Δ* mutants remained viable but had to acquire secondary mutations in order to be able to propagate. This was verified for several independent *ptk2Δ* mutants that contained suppressor mutations in the candidate gene *PMA1* and grew well after restreaking. We conclude that *PTK2* is an essential gene in *C. albicans* wild-type strains (at least in the reference strain SC5314), because it is required for the activity of the plasma membrane ATPase Pma1. Other mutations that support growth of cells lacking Ptk2 may also rescue Pma1 activity to some degree, but this was not further investigated. Blan-kenship *et al*. generated viable *ptk2* transposon insertion mutants by the UAU method, but the transposon was inserted behind the kinase domain, which might also explain the viability of these mutants [6].

In addition to firmly establishing that the uncharacterized *orf19.5376* and *orf19.3456* are indeed essential for viablity in the *C. albicans* wild-type strain SC5314, the analysis of the terminal phenotypes of the induced deletion mutants also

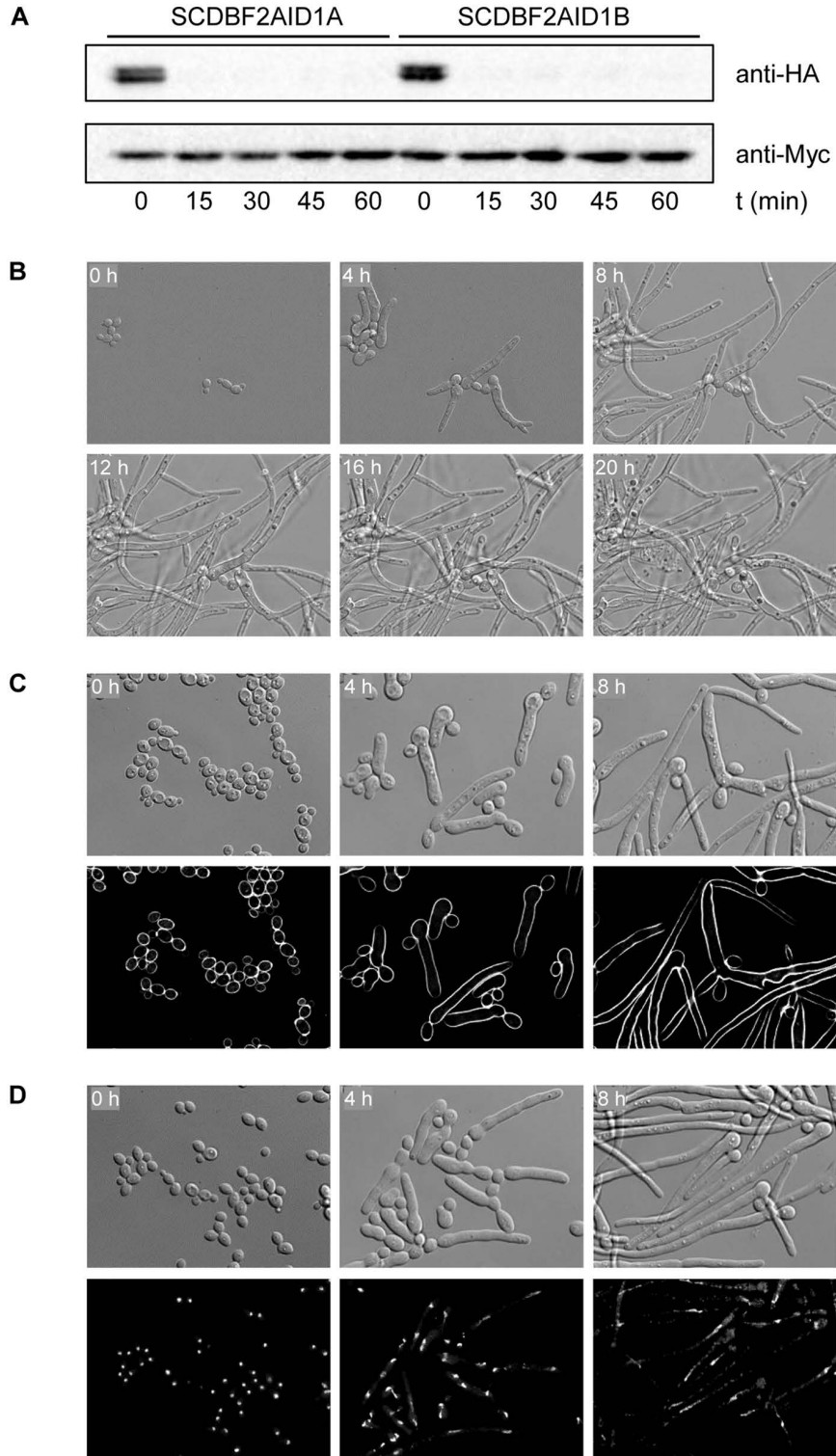

**Fig 8. Phenotypic consequences of auxin-induced Dbf2 depletion. (A)** Strains containing a single *DBF2* copy fused to the AID* cassette were grown to log phase in YPD medium. Samples were taken before and at the indicated time points after addition of 1 μM 5-Ad-IAA and analyzed by Western blotting with anti-HA and anti-Myc antibodies. **(B)** YPD-grown cells containing the degron-tagged Dbf2 were transfered to a 35-mm culture dish, covered with YPD + 1 μM 5-Ad-IAA agar, and incubated at 30°C. Images were taken every 5 min for 24 h with a Leica DMI6000 microscope (S12 Video).

The figure shows photographs of the cells at the indicated time points. **(C, D)** A YPD overnight culture of the auxin-inducible *dbf2* mutants was diluted 1:100 in YPD + 1 µM 5-Ad-IAA and grown at 30°C. Aliquots of the culture were taken every 2 h and fixed with formaldehyde. Cells were washed with PBS, stained with calcofluor white (C) or DAPI **(D)**, and imaged by DIC and fluorescence microscopy. The figure shows photographs of the cells at the indicated time points.

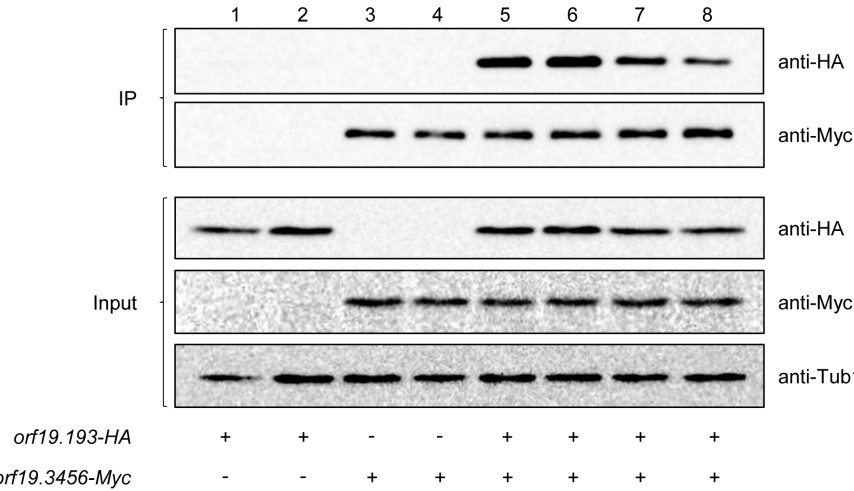

**Fig 9. The *orf19.193*-encoded protein specifically interacts with the *orf19.3456*-encoded kinase.** Strains with or without Myc-tagged *orf19.3456* and HA-tagged *orf19.193* were grown to log phase in YPD at 30°C. Protein extracts were prepared as described in materials and methods and immuno-precipitated with an anti-Myc antibody. The immunoprecipitated *orf19.3456*-encoded kinase was detected by Western blotting with an anti-Myc antibody, and the co-immunoprecipitated *orf19.193*-encoded protein with an anti-HA antibody (top panels, IP). Tagged proteins in the input samples were detected with anti-HA and anti-Myc antibodies (bottom panels, input). Detection of tubulin with an anti-tubulin antibody served as loading control for the input samples. Both independently generated series of strains were used for the experiment.

provided strong evidence that the encoded kinases are functional homologs of *S. cerevisiae* Elm1 and *S. pombe* Sid1, and we suggest to rename these ORFs *ELM1* and *SID1* (and *orf19.193 SRS1*, for Sid1 regulatory subunit). In the case of *orf19.5376* this may have seemed likely, because the encoded kinase is the closest Elm1 homolog in *C. albicans*. However, unlike Elm1 in *S. cerevisiae*, the kinase encoded by *orf19.5376* has no detectable role in Snf1 activation, because it cannot compensate for the loss of Sak1 in *C. albicans* [3]. Furthermore, several previous studies have also indicated that *orf19.5376* is essential in *C. albicans* [3,6,9,14], while *S. cerevisiae elm1Δ* mutants are viable. The phenotype of *orf19.5376Δ* mutants, which was mirrored by auxin-induced protein depletion, demonstrates that the encoded kinase has retained the known functions of Elm1 in septin organization and cytokinesis, and its loss is lethal in *C. albicans*, indicating that it cannot at least partially be substituted by other kinases involved in these processes. Interestingly, *ELM1* is not essential in *C. auris* and its deletion in several different strains resulted in constitutive pseudohyphal growth [40,41], similar to the phenotype of some *S. cerevisiae elm1Δ* mutants and the *C. albicans orf19.5376* mutants.

A discovery that came out of our analysis of essential kinase mutants is that *C. albicans* possesses a SIN pathway comprising the three kinases Cdc15, Sid1, and Dbf2 instead of a MEN pathway with only two kinases, Cdc15 and Dbf2, as was previously thought. While the direct relationships between these kinases have not yet been experimentally addressed in *C. albicans*, the lethal cytokinesis defect of all mutants lacking one of the three kinases indicates that Cdc15 cannot bypass Sid1 for activation of the downstream kinase Dbf2. One notable difference between the phenotypes of previously reported *tem1* and *cdc15* knock-down mutants as compared with our *orf19.3456* mutants is that the former showed a mitotic arrest, resulting in cells with two nuclei that did not enter subsequent rounds of nuclear division [36,39].

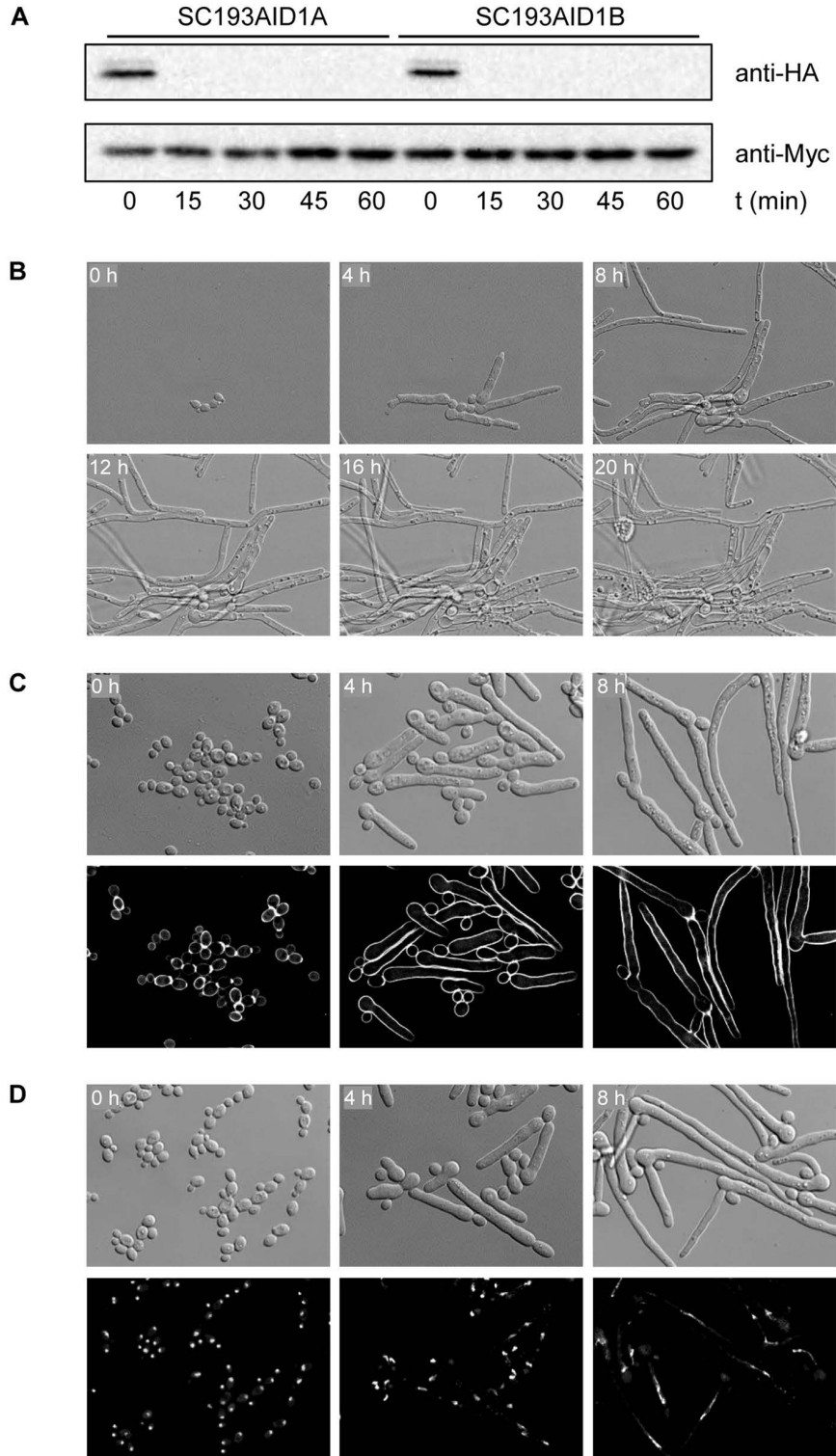

**Fig 10. Phenotypic consequences of auxin-induced degradation of the *orf19.193*-encoded protein. (A)** Strains containing a single *orf19.193* copy fused to the AID* cassette were grown to log phase in YPD medium. Samples were taken before and at the indicated time points after addition of 1 μM 5-Ad-IAA and analyzed by Western blotting with anti-HA and anti-Myc antibodies. **(B)** YPD-grown cells containing the degron-tagged *orf19.193* were transfered to a 35-mm culture dish, covered with YPD + 1 μM 5-Ad-IAA agar, and incubated at 30°C. Images were taken every 5 min for 24 h with

a Leica DMI6000 microscope (S14 Video). The figure shows photographs of the cells at the indicated time points. **(C, D)** A YPD overnight culture of the auxin-inducible *orf19.193* mutants was diluted 1:100 in YPD + 1 μM 5-Ad-IAA and grown at 30°C. Aliquots of the culture were taken every 2 h and fixed with formaldehyde. Cells were washed with PBS, stained with calcofluor white (C) or DAPI **(D)**, and imaged by DIC and fluorescence microscopy. The figure shows photographs of the cells at the indicated time points.

In contrast, we observed multinucleate cells in *orf19.3456* mutants, similar to what has been described for *S. pombe sid1* mutants [31]. However, the nuclear bodies in *orf19.3456* mutant hyphae were often extended, pointing to a mitotic delay from which some cells eventually could escape (Figs 5, S11 Fig). While it is possible that loss of Tem1 and Cdc15 result in a stronger defect in mitotic exit than does loss of Sid1 and Dbf2, the phenotypic differences may also be due to the fact that depletion of Tem1 and Cdc15 required 6 h and 12 h, and hyphal growth began only at 8 h and 20 h, respectively, after promoter shut-off in the repressible mutants [36,39], whereas auxin-induced protein depletion occurred within minutes in our mutants and hyphal growth started much earlier than in the *tem1* and *cdc15* knock-down mutants (Figs 7, 8, 10; S10 Video, S12 Video, S14 Video), which may have allowed detection of additional nuclear division events. Cells with three or more nuclei have also been observed in *dbf2* knock-down mutants [38].

Differences between the MEN in *S. cerevisiae* and its counterpart in *C. albicans* have been noted before, as the terminal phosphatase Cdc14, which is essential in *S. cerevisae* and required for inactivation of the mitotic CDKs, is not essential in *C. albicans* and its primary role is the induction of cell separation after cytokinesis [37]. Furthermore, in *C. albicans* Cdc14 is degraded at the end of mitosis instead of being sequestered into the nucleolus [37]. Dbf2 has been identified as a Cdc14 interaction partner in *C. albicans* [42], indicating that Dbf2 might phosphorylate Cdc14 to promote its release from the nucleus into the cytoplasm, as in *S. cerevisiae* [43]. However, the SIN must have additional targets, because all three core kinases as well as the upstream GTPase Tem1 are essential for viability, whereas Cdc14 is dispensable.

The SIN pathway regulates cytokinesis also in filamentous fungi, although its precise functions differ from that in *S. pombe* [44–46]. In *Neurospora crassa*, phosphorylation of the Sid2 homolog DBF-2 by SID-1 has been demonstrated, indicating that Dbf2/Sid2 homologs are phosphorylated by Sid1 homologs instead of homologs of the upstream kinase Cdc15/Cdc7 in fungi with a tripartite kinase cascade [45]. The absence of a known Sid1 homolog in budding yeasts as opposed to fission yeast and filamentous fungi had been attributed to their morphological differences, but a Sid1 homolog has recently been described also in the methylotrophic yeast *Ogataea polymorpha* [47]. Our finding that *orf19.3456* encodes a functional Sid1 homolog, combined with the fact that *orf19.3456* orthologs are present in other *Candida* and related species (http://www.candidagenome.org), supports the conclusion by Maekawa et al. [47] that the SIN pathway is the ancestral type and Sid1 homologs have been lost in the *S. cerevisiae* lineage.

## Materials and methods

### Strains and growth conditions

The *C. albicans* strains used in this study are listed in S1 Table. All strains were stored as frozen stocks with 17.2% (v/v) glycerol at -80°C and subcultured on YPD agar plates (10 g yeast extract, 20 g peptone, 20 g glucose, 15 g agar per liter) at 30°C. Strains were routinely grown in YPD liquid medium at 30°C in a shaking incubator. For the selection of transformants, 200 μg/ml nourseothricin (Werner Bioagents, Jena, Germany) or 1 mg/ml hygromycin B was added to YPD agar plates. To obtain nourseothricin-sensitive derivatives in which the *SAT1* flipper cassette was excised by FLP-mediated recombination, transformants were grown overnight in YCB-BSA-YE medium (23.4 g yeast carbon base, 4 g bovine serum albumin, 2 g yeast extract per liter, pH 4.0) without selective pressure to induce the *SAP2* promoter controlling *caFLP* expression. Appropriate dilutions were plated on YPD agar plates and grown for 2 days at 30°C. Individual colonies were picked and streaked on YPD plates as well as on YPD plates with 100 μg/ml nourseothricin to confirm sensitivity. Specific growth conditions for particular experiments are described in the text and figure legends.

## Strain constructions

*C. albicans* strains were transformed by electroporation as described previously [48]. To generate FLP-deletable cassettes containing functional copies of the protein kinase genes, the coding region and flanking sequences of the genes were amplified from genomic DNA of strain SC5314 with the primers listed in S4 Table and substituted for the *SNF1* gene in the previously described pSNF1ex3 [11], resulting in plasmids pBUD32ex3, pCTK1ex3, pKSP1ex3, pPTK2ex3, pRIO1ex3, pRIO2ex3, p3456ex3, and p5376ex3. The cassettes were excised from the vector backbone and integrated at the *ADH1* locus in the corresponding heterozygous M2 mutants [2] to obtain M3 mutants (see Fig 1 and S1 Table). The previously described gene deletion cassettes [2,3] were then used to delete the second endogenous alleles in these strains to generate M4 mutants. These strains were grown overnight in YCB-BSA-YE medium to induce FLP-mediated excision of the *SAT1* flipper cassette, resulting in M5 mutants which contained only the FLP-deletable gene copy. The insert from plasmid pSAP2FL1 [11] was used to integrate the *ecaFLP* gene under control of the *SAP2-1* promoter into the *SAP2-1* allele of M5 and M3 mutants to generate the conditional M6 mutants and the M7 control strains, respectively. Induction of *ecaFLP* expression by passage in YCB-BSA-YE medium resulted in the loss of the FLP-deletable gene copy. When the null mutants were viable (*bud32Δ*, *ctk1Δ*, *rio1Δ*, *rio2Δ*), corresponding M8 mutants (from M6) and M9 control strains (from M7) were retained. An exception were the *KSP1* mutants for which two types of derivatives of the M4 strains were obtained. In some clones, both the *SAT1* flipper cassette and the FLP-deletable ectopic *KSP1* copy were deleted, and these were kept as *ksp1Δ* null mutants (SCKSP1M5). Others had excised only the *SAT1* flipper cassette but retained the ectopic *KSP1* copy, and these were used as control strains (SCKSP1M6).

The kinase-dead *orf19.3456*[K41R] allele was amplified in two parts from genomic DNA of strain SC5314 with the primer pairs 3456.01/3456.07 and 3456.06/3456.05, followed by a fusion PCR with primers 3456.01 and 3456.05; the overlapping primers 3456.07 and 3456.06 changed the lysine codon AAA (positions +121 to +123 in *orf19.3456*) into the arginine codon AGA. The PCR product was digested with SacI/SacII and substituted for the upstream flanking sequence in the *orf19.3456* deletion cassette to obtain p3456[K41R]. The kinase-dead *orf19.5376*[K65R] allele was generated in analogous fashion with the primer pairs 5376.01/5376.13 and 5376.12/5376.11, thereby changing the lysine codon AAA (positions +193 to +195 in *orf19.5376*) into the arginine codon AGA. The PCR product was digested with SacI/SacII and substituted for the upstream flanking sequence in the *orf19.5376* deletion cassette to obtain p5376[K65R]. The inserts from these plasmids were integrated into one of the inactivated endogenous alleles of the corresponding M5 mutants to generate M8 mutants, followed by recycling of the *SAT1* flipper cassette to produce M9 mutants. The insert from plasmid pSAP2FL1 was then used to integrate the *ecaFLP* gene under control of the *SAP2-1* promoter into the *SAP2-1* allele to generate M10 mutants.

To construct 3xHA-tagged *orf19.3456* and *orf19.3456*[K41R] alleles, their upstream and coding sequences were amplified with primers 3456.01 and 3456.08; the latter primer introduced a KasI site, encoding a Gly-Ala linker, instead of the stop codon. The PCR products were digested with SacI/KasI and cloned together with a KasI-SacII fragment from pMIG1H3 [17], encoding three copies of the HA epitope followed by a stop codon and the *ACT1* transcription termination sequence, into the SacI/SacII-digested p3456[K41R] to obtain p3456H1 and p3456H[K41R], respectively. 3xHA-tagged *orf19.5376* and *orf19.5376*[K65R] alleles were generated in an analogous way using primers 5376.01 and 5376.14 to obtain plasmids p5376H1 and p5376H[K65R]. The inserts from these plasmids were integrated into one of the inactivated endogenous alleles of the corresponding M5 mutants to generate M11 and M13 mutants, followed by recycling of the *SAT1* flipper cassette to produce M12 and M14 mutants. The inserts from p3456H1 and p5376H1 were also inserted into the remaining wild-type *orf19.3456* and *orf19.5376* alleles of the M2 mutants, followed by recycling of the *SAT1* flipper cassette to generate H2 strains expressing a single 3xHA-tagged copy these genes from their own promoter. Subsequent integration of the 3xHA-tagged *orf19.3456* into the already deleted second allele resulted in H4 strains.

To generate an *orf19.193* deletion cassette, the *orf19.193* upstream and downstream regions were amplified with the primer pairs 193.09/193/10 and 193.11/193.12, respectively, and cloned on both sides of the *SAT1* flipper cassette in

plasmid pSFS5 [5]. Replacement of one of the *orf19.193* alleles in strain SC5314 by the *orf19.193* deletion cassette, followed by recycling of the *SAT1* flipper, generated heterozygous M2 mutants.

To obtain auxin-inducible *orf19.3456* and *orf19.5376* mutants, the AID* cassette from plasmid pHLP728 (a kind gift from Mark Hall) was amplified in two parts with the primer pairs AID01/AID02 and AID03/AID04. The PCR products were digested with KasI/NsiI and NsiI/XhoI, respectively, and, after fusion at the introduced NsiI sites, inserted into the KasI/XhoI-digested p3456H1 and p5376H1 to generate p3456AID1 and p5376AID1, respectively. The inserts from these plasmids were used to tag the remaining wild-type *orf19.3456* and *orf19.5376* alleles in the corresponding M2 mutants with the AID* cassette. The resulting SC3456AID1 and SC5376AID1 strains express single *orf19.3456* and *orf19.5376* alleles, respectively, containing a C-terminal fusion to the auxin-inducible degron, the *OsTIR1* gene under control of the *C. albicans ACT1* promoter, and a nourseothricin resistance marker [a modified version of the *caSAT1* marker generated in the Hall lab [28]]. Auxin-inducible *dbf2* and *orf19.193* mutants were generated in an analogous fashion. The 3' part of the *DBF2* coding region was amplified with primers DBF2.12 and DBF2.06 and the PCR product digested with SacI/KasI and cloned together with the KasI-XhoI fragment from p3456AID1 containing the AID* cassette into the SacI/XhoI-digested pDBF2M1 [2] to generate pDBF2AID1. The 3' part of *orf19.193* was amplified with primers 193.13/193.02 and the *orf19.193* downstream region with primers 193.11 and 193.12. The PCR products were digested with SacI/KasI and XhoI/ApaI, respectively, and substituted for the *orf19.3456* flanking sequences in p3456AID1 to obtain p193AID1. The inserts from the latter plasmids were used to tag the remaining wild-type *DBF2* and *orf19.193* alleles in the corresponding M2 mutants with the AID* cassette.

To facilitate HA-, Myc- and GFP-tagging in nourseothricin- or hygromycin-resistant strains, the *HygB* marker from plasmid pGRP2M2 [49] was first amplified in two parts with the primer pairs HygB-1/HygB-8 and HygB-9/HygB-2, followed by a fusion PCR with primers HygB-1 and HygB-2. The complementary primers HygB-8 and HygB-9 introduced a silent T285G substitution that removed an internal SacI site. The PCR product was digested with XhoI/PstI and cloned in the vector pBluescript II KS to obtain plasmid pHygB. A fragment comprising a KasI linker, three tandem copies of the Myc epitope sequence followed by a stop codon, and the *ACT1* transcription termination sequence was amplified from plasmid pKIS1Myc3 [50] with primers Myc3KasI and ACT1TSalI, digested with ApaI/SalI, and cloned in the ApaI/XhoI-digested pHygB to generate pMyc3-HygB. Furthermore, a shorter version of the *caSAT1* marker (without the *ACT1* intron) was obtained by amplifying the *caSAT1* gene in two parts with the primer pairs ACT1P1/ACT1P2 and SAT9/SAT10, followed by a fusion PCR with primers ACT1P1 and SAT10. The PCR product was digested with XhoI/PstI and cloned in the vector pBluescript II KS to generate pSAT5 with the modified *caSAT2* marker. A fragment comprising a KasI linker, three tandem copies of the Myc epitope sequence followed by a stop codon, and the *ACT1* transcription termination sequence was amplified as described above and cloned in the ApaI/XhoI-digested pSAT5 to generate pMyc3-SAT2.

To generate a 3xMyc-tagged *orf19.3456*, the *orf19.3456* downstream region was amplified with primers 3456.09 and 3456.04. The PCR product was digested with PstI/ApaI and cloned together with the KasI/PstI digested *3xMyc-ACT1T-caSAT2* fragment from pMyc3-SAT2 into the KasI/ApaI-digested p3456H1. The insert from this plasmid was used to replace the remaining *orf19.3456* wild-type allele in the heterozygous M2 mutants as well as one of the *orf19.3456* wild-type alleles in the strains containing a single HA-tagged *orf19.193* allele (described below). To generate a 3xMyc-tagged *orf19.193*, a part of the *orf19.193* coding sequence was amplified with the primers 193.01 and 193.02; the latter primer introduced a KasI site, encoding a Gly-Ala linker, instead of the stop codon. The *orf19.193* downstream region was amplified with primers 193.03 and 193.04. The PCR products were digested with ApaI/KasI and PstI/SacI, respectively, and cloned together with the KasI-PstI fragment from pMyc3-HygB in the ApaI/SacI-digested vector pBluescript II KS to generate p193Myc3. A *3xHA-ACT1T* fragment was amplified from pMIG1H3 with the primers MIG1HAfwd and ACT1TSalI, digested with KasI/SalI, and cloned together with the XhoI-PstI *HygB*-Fragment from pHygB in the Kas/PstI-digested p193Myc3 to obtain p193H1, thereby exchanging the 3xMyc tag for a 3xHA tag. The insert from p193H1 was used to replace the remaining *orf19.193* wild-type allele in the heterozygous M2 mutants as well as one of the *orf19.193* wild-type alleles in the strains containing a single Myc-tagged *orf19.3456* allele.

For *GFP*-tagging, a fragment containing *GFP* and the *ACT1* transcription termination sequence was amplified from plasmid pYOR1G1 [51] with the primers GFP28 and ACT1TSalI. The PCR product was digested with KasI/SalI and used to replace the 3xMyc tag in pMyc3-HygB by *GFP*, yielding pGFP-HygB. To construct a *CDC3-GFP* fusion, a part of the *CDC3* coding sequence was amplified with the primers CDC3.01 and CDC3.02; the latter primer introduced a KasI site, encoding a Gly-Ala linker, instead of the stop codon. The *CDC3* downstream region was amplified with primers CDC3.03 and CDC3.04. The PCR products were digested with ApaI/KasI and PstI/SacI, respectively, and cloned together with the KasI-PstI fragment from pGFP-HygB in the vector pBluescript II KS to generate pCDC3G1. The insert from this plasmid was used to replace one of the endogenous *CDC3* alleles by a *GFP*-tagged copy in the wild type and the auxin-inducible *orf19.3456* and *orf19.5376* mutants. In all cases two independent series of mutants (A and B) were generated (see S1 Table).

### Isolation of genomic DNA and Southern hybridization

Genomic DNA from *C. albicans* strains was isolated as described previously [4]. The DNA was digested with appropriate restriction enzymes, separated on a 1% agarose gel, transferred by vacuum blotting onto a nylon membrane, and fixed by UV crosslinking. Southern hybridization with enhanced chemiluminescence-labeled probes was performed with the Amersham ECL Direct Nucleic Acid Labelling and Detection System (Cytiva) according to the instructions of the manufacturer.

### Determination of induced gene deletion efficiency

Conditional deletion mutants and control strains were grown overnight at 30°C in YCB-BSA-YE medium to induce *ecaFLP* expression from the *SAP2* promoter. Tenfold dilution series were prepared and the $10^{-2}$ dilutions used to determine the total number of cells/ml in a counting chamber (Thoma neu). The number of viable cells (CFUs) was determined by plating 100 µl of the $10^{-3}$ dilutions of the conditional mutants and 100 µl of the $10^{-6}$ dilutions of the control strains on YPD plates. The number of colonies was determined after two days of growth at 30°C.

### Time lapse microscopy

YPD overnight cultures of the strains were diluted 1:5 in water. Five µl of the diluted cell suspension was transferred to an IBIDI µ-Dish 35 mm and covered with a previously prepared YPD agar pad, without or with 1 µM 5-Ad-IAA (TCI). The µ-Dish was placed in a Leica DMI6000 microscope equipped with a climate chamber, prewarmed to an ambient temperature of 30°C. DIC images were acquired at 5 min intervals for 24 h. Image processing was carried out with FIJI software [52].

### Fluorescence microscopy

For cell wall and nuclei staining, cells from liquid cultures were fixed with 4% formaldehyde and washed with PBS. Fixed cells were stained with 1 µg/ml calcofluor white or 5 µg/ml DAPI. For localization of GFP-tagged Cdc3, cells were fixed with 4% paraformaldehyde, washed with PBS, and stained with 1 µg/ml calcofluor white or 5 µg/ml DAPI. Cells were imaged with a Leica DMI6000 microscope using appropriate filters for fluorescence detection. Z-stacks were acquired over 5 µm in 0.5 µm increments and deconvolved using LAS X software. Additional image processing was carried out with FIJI software [52].

### Western blotting

Overnight cultures of the strains were diluted to an $OD_{600}$ of 0.4 in fresh YPD and grown for 5 h at 30°C. Cells were collected by centrifugation, washed with ice-cold water, and resuspended in 300 µl breaking buffer (50 mM Tris-HCl pH 8, 250 mM NaCl, 5 mM EDTA, 0.1% [v/v] Triton X-100, cOmplete EDTA-free Protease Inhibitor Cocktail and PhosStop

Phosphatase Inhibitor Cocktail [Roche]). An equal volume of 0.5 mm acid-washed glass beads was added to each tube. Cells were mechanically disrupted on a FastPrep-24 cell-homogenizer (MP Biomedicals) with three 40 s pulses, with 5 min on ice between each pulse. Cell lysates were centrifuged at 21,000 x $g$ for 15 min at 4°C, the supernatant was collected, and the protein concentration was quantified using the Bradford protein assay. Equal amounts of protein of each sample were mixed with one volume of 2 x Laemmli buffer, heated for 5 min at 95°C, and separated on an SDS-polyacrylamide (8% or 10%) gel. Separated proteins were transferred onto a nitrocellulose membrane with a mini-Protean System (Bio-Rad). To detect HA-tagged proteins, membranes were blocked with 5% milk in TBST and incubated overnight with rat monoclonal anti-HA-peroxidase antibody, clone 3F10 (Roche). For the detection of tubulin, membranes were blocked with 5% milk in TBST and incubated overnight at 4°C with rat anti-tubulin alpha antibody MCA 78G (Bio-Rad), washed with TBST, and then incubated with rabbit anti-rat HRP-conjugated antibody STAR21B (Bio-Rad). Myc-tagged proteins were detected with anti-Myc (71D10) rabbit mAb (Cell Signaling Technology) and anti-rabbit HRP G-21234 (Invitrogen) as secondary antibody. To reprobe the immunoblots, membranes were incubated in stripping buffer (0.2 M glycine, 0.1% SDS, 1% Tween 20, pH 2.2), and washed in PBS and TBST before blocking with 5% milk. Signals were generated with the ECL chemiluminescence detection system (Cytiva) and captured with the ImageQuant LAS 4000 imaging system (Cytiva).

## Co-immunoprecipitation

For Co-IP of proteins with the HA-tagged *orf19.3456*-encoded kinase, cells were grown to log phase in YPD and treated for 1 h with 1 mM DSP (CAS 57757-57-0) at room temperature. Crosslinking was stopped by adding Tris-HCl pH 7.4 to a final concentration of 50 mM and incubating for additional 15 min. The cells were then washed with ice-cold water and frozen in liquid nitrogen. Cell extracts were obtained by grinding the cells using mortar and pestle under liquid nitrogen. The powdered extracts were mixed with two volumes of IP-Buffer (40 mM Tris-HCl pH 7.4, 250 mM sodium citrate, 150 mM NaCl, 1% Triton X-100) and centrifuged for 15 min (21,000 x $g$, 4°C). The supernatants were recovered, mixed with 100 µl anti-HA agarose beads (Pierce 26181, Thermo Scientific), and incubated for 1 h at 4°C with end-over-end rotation. The agarose beads were washed with IP-buffer, followed by three washes with PBS and a final wash with ultrapure water. Bound proteins were eluted with 50 µl 0.1% (v/v) TFA in 30% (v/v) aqueous acetonitrile. The elution step was repeated two additional times and the eluates were combined for a final volume of 150 µl. 10 µl of the eluates were dried in a SpeedVac, resuspended in 2x protein sample buffer and analyzed by Western blotting using an anti-HA HRP-conjugated antibody. Three biological replicates of the tagged strains (two from the A strain and one from the B strain) and three biological replicates of the untagged control strain SC5314 were used for proteomic analysis.

For specific Co-IP of the HA-tagged *orf19.193*-encoded protein with the Myc-tagged *orf19.3456*-encoded kinase, cell extracts were prepared as described above. Supernatants were mixed with 100 µl anti-c-Myc agarose beads (Pierce 20168, Thermo Scientific) and incubated for 18 h at 4°C with end-over-end rotation. The agarose beads were washed with IP-buffer, followed by three washes with PBS, and bound proteins eluted with 50 µl of 2x protein sample buffer. Immuno-precipitated samples and corresponding input samples were analyzed by Western blotting as described above.

## Proteomics

**Tryptic digestion.** Co-IP eluates (35 µl of 0.1% TFA in 30/70 ACN/$H_2O$, v/v) were evaporated to dryness in a vacuum concentrator (Eppendorf). Proteins were resolubilized in 50 µl of 50 mM triethylammonium bicarbonate (TEAB) in 50/50 trifluoroethanol (TFE)/$H_2O$ (v/v). Cysteine thiols were reduced and carbamidomethylated in one step for 30 min at 70°C by addition of each 1 µL of 500 mM TCEP (tris(2-carboxyethyl)phosphine) and 625 mM 2-chloroacetamide (CAA) per 50 µl sample. Samples were again evaporated to dryness and resolubilized in 50 µl of 100 mM TEAB in 5/95 TFE/$H_2O$ (v/v). Proteins were digested for 18 h at 37°C after addition of 1 µl of a 2 µg/µl Trypsin/Lys-C mix (in 50 mM acetic acid). Tryptic peptides were evaporated to dryness with a vacuum concentrator. Dried peptides were resolubilized in 30 µl 0.05% TFA and 2% ACN in water by pipetting up and down several times followed by 15 min ultrasonic bath treatment and Vortex

homogenization. Finally, the peptides were filtered through 0.2 µm Ultrafree-MC hydrophilic PTFE filters (Merck-Millipore) for 15 min at 16,000 × $g$ (8°C). The filtrate was transferred to HPLC vials.

**LC-MS/MS analysis.** Injection volume was 6 µl. LC-MS/MS analysis was performed on an Ultimate 3000 nano RSLC system connected to a Orbitrap Exploris 480 mass spectrometer (both Thermo Fisher Scientific, Waltham, MA, USA) with FAIMS. Peptide trapping for 5 min on an Acclaim Pep Map 100 column (2 cm x 75 µm, 3 µm) at 5 µL/min was followed by separation on a µPACneo 110 column. Mobile phase gradient elution of eluent A (0.1% [v/v] formic acid in water) mixed with eluent B (0.1% [v/v] formic acid in 90/10 acetonitrile/water) was performed using the following gradient: 0 min at 4% B and 750 nl/min, 10 min at 9% B and 750 nl/min, 12 min at 9.5% B and 300 nl/min, 55 min at 25% B and 300 nl/min, 70 min at 50% B and 300 nl/min, 75 min at 96% B and 300 nl/min, 78–80 min at 96% B and 750 nl/min, 80.1-90 min at 4% B and 750 nl/min. Positively charged ions were generated at spray voltage of 2.2 kV using a stainless steel emitter attached to the Nanospray Flex Ion Source (Thermo Fisher Scientific). The quadrupole/orbitrap instrument was operated in Full MS/ data-dependent MS2 mode. Precursor ions were monitored at m/z 300–1100 at a resolution of 120,000 FWHM (full width at half maximum) using a maximum injection time (ITmax) of 50 ms and 300% normalized AGC (automatic gain control) target. Precursor ions with a charge state of z = 2–5 were filtered at an isolation width of $m/z$ 4.0 amu for further fragmentation at 28% HCD collision energy. MS2 ions were scanned at 15,000 FWHM (ITmax = 40 ms, AGC = 200%). Each sample was measured in triplicate with a different compensation voltage (-42 V, -57 V, -72 V).

**Protein database search.** Tandem mass spectra were searched against the UniProt database of *Candida albicans* SC5314 (https://www.uniprot.org/proteomes/UP000000559; 2025/10/20) using FragPipe 23.1 and the database search algorithm MS Fragger 4.3. Two missed cleavages were allowed for the tryptic digestion. The precursor mass tolerance was set to 10 ppm and the fragment mass tolerance was set to 20 ppm. Modifications were defined as dynamic Met oxidation, protein N-terminal acetylation, and Ser/Thr/Tyr phosphorylation (Sequest HT only). DDA+was used for data-dependent acquisition with wide window isolation and chimeric spectra detection of the top 5 precursor ions. MS Booster with the deep learning model of DIA-NN for retention time and mass spectra prediction was applied together with Percolator and a reverse decoy database for q-value validation of spectral matches using a strict false discovery rate (FDR) <1% on both peptide and protein level. Label-free protein quantification was based on the IonQuant 1.11.11 algorithm. Normalization was performed by using the total protein amount method per sample group. Imputation of missing quan values was applied by using deterministic random abundance values of 50–100% of the lowest abundance identified per sample. Enriched proteins were defined as a fold change of >2, pvalue <0.05 and at least identified in 2 of 3 biological replicates of the HA-tagged sample group. Statistics and data visualization was performed with R 4.5.1 and RStudio 2025.09.0.

## Supporting information

**S1 Fig. Growth of the conditional mutants and control strains.** The conditional mutants (M6), which contain only the ectopically integrated gene copy, and control strains (M7), which additionally retain one of the endogenous alleles (M3 in the case of *KSP1*) were streaked on YPD plates and incubated for 2 days at 30°C. Both independently generated strain series are shown in each case. WT, parental wild-type strain SC5314.
(PDF)

**S2 Fig. Growth of viable *pekΔ* null mutants.** The M8 null mutants (M5 in the case of *KSP1*) and M9 control strains (M6 in the case of *KSP1*) were streaked on YPD plates and incubated at 30°C and at 37°C. Photographs were taken after 2, 4, and 6 days. Both independently generated strain series are shown in each case.
(PDF)

**S3 Fig. Analysis of *ptk2Δ* suppressor mutants.** (A) Four viable clones obtained after the induced gene deletion in the conditional M6 mutants (two each from strains A and B) were streaked on YPD plates and incubated for two days at 30°C.

The wild-type strain SC5314 (WT) is shown for comparison. (B) Southern hybridization analysis of ClaI-digested genomic DNA of the wild-type strain SC5314, the conditional M6 mutants, and the four *ptk2Δ* suppressor mutants with probes from the *PTK2* downstream and coding regions confirms the absence of *PTK2* in the suppressor mutants. (C) Southern hybridization analysis of XhoI/SpeI-digested genomic DNA of the same strains with an *ADH1* upstream probe demonstrates correct FLP-mediated excision of the ectopically integrated *PTK2* copy in the suppressor mutants. A labeled size marker (M, in kb) was included in the probes. The identities of the hybridizing fragments are indicated.
(PDF)

**S4 Fig. Growth of *orf19.3456Δ* and *orf19.5376Δ* control strains.** The M7 mutants were passaged overnight in YCB-BSA-YE medium to induce FLP-mediated excison of the ectopically integrated gene copy. The cultures were diluted in water, transferred to a 35 mm culture dish, covered with YPD agar, and incubated at 30°C. Images were taken every 5 min with a DMI6000 Leica inverted microscope (S4-S5 Videos). The figure shows photographs of the cells at the indicated time points.
(PDF)

**S5 Fig. Chitin and nuclei staining of serum-induced wild-type hyphae.** A YPD overnight culture of the wild-type strain SC5314 was diluted 1:100 in YPD+10% FCS and incubated for 6 h at 37°C. Aliquots of the culture were taken every 2 hours and fixed with formaldehyde. Cells were washed with PBS and stained with calcofluor white (A) or DAPI (B). Cells were imaged by DIC and fluorescence microscopy.
(PDF)

**S6 Fig. Chitin and nuclei staining of *orf19.3456Δ* and *orf19.5376Δ* control strains.** YCB-BSA-YE overnight cultures of the M7 mutants were diluted 1:100 in YPD medium and grown at 30°C. Aliquots of the cultures were taken after 4 h and fixed with formaldehyde. Cells were washed with PBS and stained with calcofluor white (A and C) or DAPI (B and D). Cells were imaged by DIC and fluorescence microscopy.
(PDF)

**S7 Fig. Analysis of *orf19.3456* and *orf19.5376* kinase-dead mutants.** (A) Viability of strains that retain a kinase-dead allele at the endogenous locus after FLP-mediated excision of the ectopically integrated wild-type copy. YCB-BSA-YE overnight cultures of the conditional M10 mutants were appropriately diluted and the total number of cells and CFUs was determined as described in materials and methods. (B) Microscopic appearance of the kinase-dead mutants. Cells from the YCB-BSA-YE cultures were diluted, transferred to a culture dish, covered with YPD agar, and observed by video microscopy at 30°C (S6-S7 Videos). Pictures were taken at the indicated time points. (C) Kinase-dead proteins are produced at wild-type levels. Strains expressing an HA-tagged wild-type or kinase-dead allele from the endogenous genomic locus in addition to an ectopically integrated wild-type copy were grown to log-phase in YPD medium and analyzed by Western blotting with anti-HA and anti-tubulin antibodies. (D) HA-tagged kinases are functional. Strains containing a single wild-type (M2) or HA-tagged (H2) allele at the endogenous locus were grown for 2 days at 30°C on YPD plates. The wild-type strain SC5314 is shown for comparison. Results for two independently generated series of strains are shown in (A), (C), and (D).
(PDF)

**S8 Fig. Auxin-induced degradation of the *orf19.5376*-encoded kinase causes defects in septum formation and nuclear localization.** A YPD overnight culture of the auxin-inducible *orf19.5376* mutants was diluted 1:100 in YPD+1 μM 5-Ad-IAA and grown at 30°C. Aliquots of the culture were taken every 2 h and fixed with formaldehyde. Cells were washed with PBS, stained with calcofluor white (A) or DAPI (B), and imaged by DIC and fluorescence microscopy. The figure shows photographs of the cells at the indicated time points. Identically treated control cells (the heterozygous M2 mutants containing a single untagged *orf19.5376* allele) are shown in (C) and (D).
(PDF)

**S9 Fig. Septin localization in wild-type cells.** A YPD overnight culture of the wild-type strain SC5314 containing a *GFP*-tagged *CDC3* allele was diluted 1:100 in YPD + 1 µM 5-Ad-IAA and grown at 30°C. Aliquots of the culture were taken every 2 h and fixed with paraformaldehyde. Cells were washed with PBS, stained with DAPI, and imaged by DIC (left panels) and fluorescence microscopy (middle panels). The figure shows photographs of the cells at the indicated time points, including overlays of the DIC and fluorescence micrographs (right panels).
(PDF)

**S10 Fig. Septin localization in serum-induced wild-type hyphae.** A YPD overnight culture of the wild-type strain SC5314 containing a *GFP*-tagged *CDC3* allele was diluted 1:100 in YPD with 10% serum and grown at 37°C. Aliquots of the culture were taken every 2 h and fixed with paraformaldehyde. Cells were washed with PBS, stained with DAPI (A) or calcofluor white (B), and imaged by DIC and fluorescence microscopy. The figure shows photographs of the cells at the indicated time points, including overlays of the DIC and fluorescence micrographs.
(PDF)

**S11 Fig. Auxin-induced degradation of the *orf19.3456*-encoded kinase causes formation of aseptate, multinucleate hyphae.** A YPD overnight culture of the auxin-inducible *orf19.3456* mutants was diluted 1:100 in YPD + 1 µM 5-Ad-IAA and grown at 30°C. Aliquots of the culture were taken every 2 h and fixed with formaldehyde. Cells were washed with PBS, stained with calcofluor white (A) or DAPI (B), and imaged by DIC and fluorescence microscopy. The figure shows photographs of the cells at the indicated time points. Identically treated control cells (the heterozygous M2 mutants containing a single untagged *orf19.3456* allele) are shown in (C) and (D).
(PDF)

**S12 Fig. Chitin and nuclei staining of auxin-treated *DBF2* control strains.** A YPD overnight culture of the heterozygous M2 mutants containing a single untagged *DBF2* allele was diluted 1:100 in YPD + 1 µM 5-Ad-IAA and grown at 30°C. Aliquots of the culture were taken every 2 h and fixed with formaldehyde. Cells were washed with PBS, stained with calcofluor white (A) or DAPI (B), and imaged by DIC and fluorescence microscopy. The figure shows photographs of the cells at the indicated time points.
(PDF)

**S13 Fig. Chitin and nuclei staining of auxin-treated *orf19.193* control strains.** A YPD overnight culture of the heterozygous M2 mutants containing a single untagged *orf19.193* allele was diluted 1:100 in YPD + 1 µM 5-Ad-IAA and grown at 30°C. Aliquots of the culture were taken every 2 h and fixed with formaldehyde. Cells were washed with PBS, stained with calcofluor white (A) or DAPI (B), and imaged by DIC and fluorescence microscopy. The figure shows photographs of the cells at the indicated time points.
(PDF)

**S1 Video. Time-lapse microscopy of the wild-type strain SC5314.**
(AVI)

**S2 Video. Time-lapse microscopy of *orf19.3456Δ* mutants.**
(AVI)

**S3 Video. Time-lapse microscopy of *orf19.5376Δ* mutants.**
(AVI)

**S4 Video. Time-lapse microscopy of *orf19.3456Δ* control cells.**
(AVI)

**S5 Video. Time-lapse microscopy of orf19.5376Δ control cells.**
(AVI)

**S6 Video. Time-lapse microscopy of orf19.3456K41R mutants.**
(AVI)

**S7 Video. Time-lapse microscopy of orf19.5376K65R mutants.**
(AVI)

**S8 Video. Time-lapse microscopy of auxin-inducible orf19.5376 mutants.**
(AVI)

**S9 Video. Time-lapse microscopy of auxin-treated orf19.5376 control cells.**
(AVI)

**S10 Video. Time-lapse microscopy of auxin-inducible orf19.3456 mutants.**
(AVI)

**S11 Video. Time-lapse microscopy of auxin-treated orf19.3456 control cells.**
(AVI)

**S12 Video. Time-lapse microscopy of auxin-inducible dbf2 mutants.**
(AVI)

**S13 Video. Time-lapse microscopy of auxin-treated dbf2 control cells.**
(AVI)

**S14 Video. Time-lapse microscopy of auxin-inducible orf19.193 mutants.**
(AVI)

**S15 Video. Time-lapse microscopy of auxin-treated orf19.193 control cells.**
(AVI)

**S1 Table.** *C. albicans* **strains used in this study.**
(XLSX)

**S2 Table. Significantly enriched proteins identified after Co-IP with the *orf19.3456*-encoded kinase.**
(XLSX)

**S3 Table. Proteomics results.**
(XLSX)

**S4 Table. Oligonucleotide primers used in this study.**
(XLSX)

## Acknowledgments

We thank Mark Hall for the generous gift of plasmid pHLP728 containing the AID* cassette before publication.

## Author contributions

**Conceptualization:** Joachim Morschhäuser.

**Data curation:** Bernardo Ramírez-Zavala, Thomas Krüger, Olaf Kniemeyer, Joachim Morschhäuser.

**Formal analysis:** Bernardo Ramírez-Zavala, Thomas Krüger, Olaf Kniemeyer, Joachim Morschhäuser.

**Funding acquisition:** Olaf Kniemeyer, Joachim Morschhäuser.

**Investigation:** Bernardo Ramírez-Zavala, Ines Krüger, Sonja Schwanfelder, Johannes Lackner, Thomas Krüger, Olaf Kniemeyer, Joachim Morschhäuser.

**Methodology:** Bernardo Ramírez-Zavala, Thomas Krüger, Olaf Kniemeyer, Joachim Morschhäuser.

**Project administration:** Olaf Kniemeyer, Joachim Morschhäuser.

**Resources:** Olaf Kniemeyer, Joachim Morschhäuser.

**Supervision:** Bernardo Ramírez-Zavala, Olaf Kniemeyer, Joachim Morschhäuser.

**Validation:** Bernardo Ramírez-Zavala, Thomas Krüger, Olaf Kniemeyer, Joachim Morschhäuser.

**Visualization:** Thomas Krüger, Joachim Morschhäuser, Bernardo Ramírez-Zavala.

**Writing – original draft:** Joachim Morschhäuser.

**Writing – review & editing:** Bernardo Ramírez-Zavala, Thomas Krüger, Olaf Kniemeyer, Joachim Morschhäuser.

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
