## [Decision Letter · Decision Letter 0]

7 Dec 2025

PGENETICS-D-25-01229

Inducible gene deletion reveals essentiality of protein kinases and a septation initiation network in Candida albicans

PLOS Genetics

Dear Dr. Morschhäuser,

Thank you for submitting your manuscript to PLOS Genetics. After careful consideration, we feel that it has merit but does not fully meet PLOS Genetics's publication criteria as it currently stands. Therefore, we invite you to submit a revised version of the manuscript that addresses the points raised during the review process.

Most importantly, I agree with the concerns raised by Reviewer 2 that additional experimental evidence is needed to support the claims that orf19.3456 is a functional ortholog of S. pombe Sid1, that orf19.5376 is an ortholog of S. cerevisiae Elm1, and that orf19.193 is likely to be a regulatory subunit of orf19.193. In addition, as noted by Reviewers 2 and 3, several experiments lack the appropriate control strains for comparison, and these should be included.

We look forward to receiving your revised manuscript.

Kind regards,

Jennifer A Benanti

Academic Editor

PLOS Genetics

Geraldine Butler

Section Editor

PLOS Genetics

Aimée Dudley

Editor-in-Chief

PLOS Genetics

Anne Goriely

Editor-in-Chief

PLOS Genetics

**Journal Requirements:**

At this stage, the following Authors/Authors require contributions: Bernardo Ramírez-Zavala, Ines Krüger, Sonja Schwanfelder, Johannes Lackner, Thomas Krüger, Olaf Kniemeyer, and Joachim Morschhäuser. Please ensure that the full contributions of each author are acknowledged in the "Add/Edit/Remove Authors" section of our submission form.

The list of CRediT author contributions may be found here: https://journals.plos.org/plosgenetics/s/authorship#loc-author-contributions

4) We notice that your supplementary Figures are included in the manuscript file. Please remove them and upload them with the file type 'Supporting Information'. Please ensure that each Supporting Information file has a legend listed in the manuscript after the references list.

**Reviewers' comments:**

Reviewer's Responses to Questions

Reviewer #1: In this study, the authors apply their inducible gene deletion strategy to definitively assign function to C. albicans protein kinase genes. They extend their analysis through time course analysis, auxin-directed protein depletion, and mass spectrometry to assign pathway relationships to several essential protein kinase genes and functionally related genes. Each new essential protein kinase gene is followed up to reach clear conclusions about signaling pathways.

This study presents a truly monumental amount of work. It is of central importance to the C. albicans community because of its implications for cell biology, virulence, regulatory networks, and drug development. It will be of interest to geneticists beyond the C. albicans community because of the clarity of thought and logic, and the elegance of the approaches.

There is an additional reason that I am especially enthusiastic about the manuscript: it presents a very high level of scholarly analysis of the literature. Essentiality and protein kinase functions have been assessed in this organism for about 25 years, so most authors seem to rely simply on the accurate but more minimal gene summaries at the CGD and other databases. I appreciated especially that the authors here really dug into the original literature to explain why results have differed over the decades.

Finally, there is a quote from the authors that I have already etched into my memory and embraced in my heart. Yeah, I may even put it on a T-shirt for all to see! It is, "The occurrence of suppressor mutations that enable (normal) growth of otherwise nonviable or poorly growing mutants is indeed an issue that must be kept in mind, especially when rare homozygous mutants are obtained using traditional gene deletion methods." In one sense it is obvious, but I cannot tell you how many times authors and experimentalists forget all about this issue as soon as they get their long sought-after mutant. I really appreciate the examples that the authors drilled down into in this study.

I have a few minor criticisms for the authors' consideration.

Line 223 "because of the immediate effect of protein depletion" I thought you might say "more immediate effect."

Lines 352-354 "...we suggest to rename these ORFs ELM1 and SID1 (and orf19.193 SRS1, for Sid1 regulatory subunit)." You might put these gene names in the abstract, because many readers may overlook them here toward the end of the paper.

Table 1. A very useful summary - thank you! (No criticism.)

Table 2. I really appreciate this summary of results, but it would be more intuitive if there were an additional column listing gene name or something like that.

Reviewer #2: In this manuscript, Ramírez-Zavala et al. used an innovative inducible gene deletion strategy to assess the essentiality of eight putative essential protein kinases in the fungal pathogen Candida albicans. Of the eight kinases, PTK2, orf19.3456 and orf19.5376 were confirmed to be essential for growth under standard conditions. Under specific conditions, the authors were able to generate suppressor mutations in ptk2∆ mutants that enabled growth, namely in the Ptk2-dependent plasma membrane ATPase Pma1. Follow-up studies on the uncharacterized essential genes orf19.5376 and orf19.3456 showed the formation of filamentous cells with a defect in septum organization, which eventually lead to cell lysis. For orf19.3456, deletion mutants also exhibited multinucleated cells. To confirm the kinase activity of orf19.3456 and orf19.5376 is what was required for gene essentiality, the authors generated kinase-dead mutants, which confirmed the phenotypes of the inducible gene deletion mutants. In addition, the authors generated auxin-inducible protein depletion strains and confirmed the phenotypes of the deletion mutants. For orf19.3456, the authors performed co-IP followed by mass spectrometry to identify interacting partners, which led to the discovery of orf19.193 as a physical interactor. Depletion of degron-tagged orf19.193 resulted in the same terminal phenotype as orf19.3456 mutants, highlighting a role of this gene in C. albicans septin organization and cell cycle.

Overall, the paper presents an innovative approach to assess gene essentiality in C. albicans, emphasizing inducible gene deletion as a useful strategy to assess gene fitness in this fungal pathogen. Further, the authors establish the essentiality of several putative protein kinases that were previously uncharacterized. The authors examined the terminal phenotypes of several mutants, providing insights into the possible functions of two uncharacterized essential genes, orf19.3456 and orf19.5376. They employed complementary genetic strategies that confirmed phenotypes observed using the inducible gene deletion strategy. Nonetheless, the work could benefit from additional mechanistic insights into how orf19.3456 and orf19.5376 regulate septin organization and cytokinesis in C. albicans. To increase the impact of the manuscript, I have listed some suggestions in the point-by-point comments below:

Major comments:

1) The authors claim that orf19.3456 is the functional ortholog of S. pombe Sid1 based on cellular morphology and lack of septum formation. However, the disruption of unrelated proteins can also produce multinucleate cells and septation defects (e.g., mutants of the C. albicans RAM network). The authors should provide additional experimentation including cellular localization studies and/or functional complementation. For instance, the authors could confirm whether orf19.3456 localizes to spindle pole body in anaphase as seen with S. pombe Sid1 (Guertin et al., 2000).

2) The authors performed co-IP followed by mass spectrometry to identify orf19.193 as a physical interactor of orf9.3456. While they confirmed a orf19.193 mutant showed the same phenotypes as a orf19.3456 mutant, they were unable to validate the physical interaction co-IP coupled to western blotting. To further investigate whether orf19.193 may be a regulatory subunit of orf19.3456, the authors should perform additional experimentation; for example, cellular co-localization studies to support the mass spectrometry results.

3) The authors showed that orf19.5376 is important for septin localization and cytokinesis, which is similar to S. cerevisiae Elm1. As discussed in Comment 1, disruption of many unrelated proteins can produce these phenotypes. Thus, additional experimental evidence to support functional orthology between orf19.5376 and S. cerevisiae Elm1 is required. For example, examining the localization of orf19.5376 to confirm it is localized to the budneck. Alternatively, the authors could attempt functional complementation studies to assess whether C. albicans orf19.5376 can complement S. cerevisiae elm1 mutants and vice versa.

4) The authors employ auxin-inducible degron-tagged kinase strains in Fig 6-Fig 9, S5 Fig, and S7 Fig. For comparison, it would be helpful to include parental controls (e.g., a strain containing single wild type allele treated with 5-Ad-IAA). For comparison of filamentous cells, it is important to include controls (e.g., serum-induced filaments), which possesses normal constrictions, septin and/or nuclei as opposed to filaments induced upon cell cycle inhibition.

5) The authors employ kinase-dead orf19.3456 and orf19.5376 strains in S4 Fig. It would be helpful for comparison to include wild type controls in both viability and microscopy assays (S4 Fig A-B).

Minor comments:

1) The authors claim C. albicans orf19.3456 has no orthologue in S. cerevisiae (line 239-242). However, orf19.3456 appears to have 42.56% sequence identity to S. cerevisiae KIC1 (for reference orf19.3456 shows 40.23% identity to S. pombe SID1). Kic1 is a component of the well-established RAM network, which regulates polarized growth and cell separation. C. albicans RAM mutants exhibit cell separation defects and a multinucleate phenotype (Song et al. 2008), consistent with orf19.3456 mutant phenotypes. Thus, it should be addressed in the manuscript whether orf19.3456 may function as part of RAM network in C. albicans.

2) The authors claim that orf19.193 belongs to the same family as S. pombe Cdc14 and thus is a putative regulatory subunit of orf19.3456 (line 301-303). However, BLASTp analysis of C. albicans orf19.193 against S. pombe Cdc14 indicates poor sequence similarity (21.71%). The authors should include supporting evidence (e.g., structural similarity analysis or conserved domains search) of why they believe orf19.193 belongs to the same family as S. pombe Cdc14.

3) For ease of comparison, it would be helpful to combine Fig 6C (degron-tagged kinase strain) with S6 Fig (the auxin-treated wild type control) in the same figure.

4) Remove the delta symbol from line 197. The header should read: “orf19.5376 encodes…”

Reviewer #3: In this paper, the investigators have used a clever inducible gene deletion strategy to study a handful of kinases thought to be essential based on past inability to generate mutants in one or more reports in the human opportunistic pathogen Candida albicans. Using this strategy they found that several are actually not truly essential for growth under lab conditions, where others are. By characterizing the terminal phenotypes after gene deletion they have been able to assign a previously uncharacterized essential kinase ORF as the homolog of the S. cerevisiae ELM1 kinase involved in septin organization, bud growth and cytokinesis. For another essential uncharacterized kinase ORF, they demonstrate that it is a previously unrecognized component of the SIN pathway found in S. pombe and filamentous fungi but absent in S. cerevisiae (and thought to be generally absent in budding yeasts) and identify its putative activating subunit (also a previously uncharacterized ORF). The investigators then used a completely orthologous method, auxin-inducible protein degradation, to confirm the phenotypes and functional assignments of these two essential kinases. The application of two different methods, each producing the same phenotype, makes the study very robust and the conclusions rock solid. I find this study quite impactful as it highlights some of the problems with using permanent gene disruptions to characterize gene function and assess essentiality. In addition, the authors have provided useful new insight into the kinome of this important fungal pathogen and revealed something unexpected about a key fungal signaling pathway.

I have only a few minor comments for consideration. Otherwise, the study is well-designed and the manuscript well-written.

1. In the abstract the authors state that the product of orf19.5376 “is functionally similar” to Elm1 of S. cerevisiae. However, I think it also is the closest ortholog of Elm1 in C. albicans, correct? At least, in the Candida Genome Database Elm1 is listed as the S. cerevisiae ortholog of this kinase. I would adjust the wording to indicate this.

2. Also in the abstract I would go ahead and state the name of the S. pombe ortholog in the SIN pathway at the end (rather than just calling it the orf19.3456-encoded kinase).

3. One concern with the conclusion about the hyperpolarized “hyphal”-like structures being aseptate is that there is no control showing septa in wild-type hyphae. I understand the structures that form in these induced mutants are probably not true hyphae or pseudohyphae and that septa can be seen between some of the yeast-form cells, but it would be nice to know that the imaging method could reveal septa in the hyphal-like structures if they were present. This could be accomplished simply by inducing hyphae in the wild-type parent strain and showing the septa for comparison under identical imaging and processing conditions.

4. The authors conclude that the mutants exhibit cytokinesis defects (e.g. lines 179-180). While this may be true, cytokinesis has not been explicitly monitored. I feel this conclusion would require assessing formation and contraction of the actomyosin ring because there are many mutants in other pathways that give rise to a similar hyperpolarized cell phenotype in this organism. The results are consistent with this possibility but maybe the conclusion is worded too strong. Or the authors can provide more explanation for why they conclude cytokinesis is defective – its possible I am missing something.

**Have all data underlying the figures and results presented in the manuscript been provided?**

Large-scale datasets should be made available via a public repository as described in the *PLOS Genetics*
data availability policy, and numerical data that underlies graphs or summary statistics should be provided in spreadsheet form as supporting information., and numerical data that underlies graphs or summary statistics should be provided in spreadsheet form as supporting information., and numerical data that underlies graphs or summary statistics should be provided in spreadsheet form as supporting information., and numerical data that underlies graphs or summary statistics should be provided in spreadsheet form as supporting information.

Reviewer #1: Yes

Reviewer #2: Yes

Reviewer #3: Yes

PLOS authors have the option to publish the peer review history of their article (what does this mean?). If published, this will include your full peer review and any attached files.). If published, this will include your full peer review and any attached files.). If published, this will include your full peer review and any attached files.). If published, this will include your full peer review and any attached files.

...

Reviewer #1: No

Reviewer #2: No

Reviewer #3: No

**Figure resubmission:**
---

## [Decision Letter · Decision Letter 1]

26 Mar 2026

PGENETICS-D-25-01229R1

Inducible gene deletion reveals essentiality of protein kinases and a septation initiation network in Candida albicans

PLOS Genetics

Dear Dr. Morschhäuser,

Thank you for submitting your manuscript to PLOS Genetics. After careful consideration, we feel that it has merit but does not fully meet PLOS Genetics's publication criteria as it currently stands. Therefore, we invite you to submit a revised version of the manuscript that addresses the points raised during the review process.

As you will see, the revised manuscript has been reviewed by two of the original reviewers, both of whom are supportive of publication. Reviewer 3 has identified one minor point concerning a newly added experiment that would benefit from additional clarification prior to publication. This appears to be a simple and straightforward control that should be able to addressed easily.

Please submit your revised manuscript within by Apr 25 2026 11:59PM. If you will need more time than this to complete your revisions, please reply to this message or contact the journal office at plosgenetics@plos.org. Please include the following items when submitting your revised manuscript:

We look forward to receiving your revised manuscript.

Kind regards,

Jennifer A Benanti

Academic Editor

PLOS Genetics

Geraldine Butler

Section Editor

PLOS Genetics

Aimée Dudley

Editor-in-Chief

PLOS Genetics

Anne Goriely

Editor-in-Chief

PLOS Genetics

**Journal Requirements:**

At this stage, the following Authors/Authors require contributions: Bernardo Ramírez-Zavala, Ines Krüger, Sonja Schwanfelder, Johannes Lackner, Thomas Krüger, Olaf Kniemeyer, and Joachim Morschhäuser. Please ensure that the full contributions of each author are acknowledged in the "Add/Edit/Remove Authors" section of our submission form.

The list of CRediT author contributions may be found here: https://journals.plos.org/plosgenetics/s/authorship#loc-author-contributions

3)  Please ensure that the funders and grant numbers match between the Financial Disclosure field and the Funding Information tab in your submission form. Note that the funders must be provided in the same order in both places as well.

**Reviewers' comments:**

Reviewer's Responses to Questions

**Comments to the Authors:**

Reviewer #1: This study presents a monumental amount of rigorous, highly significant work. My comments have all been addressed. I hope to see it in press soon, so that it can benefit our community.

Reviewer #3: I still think this is a great study and the overall rigor is high and suitable for publishing in PLoS Genetics. I only had a few comments on the initial submission and most were addressed adequately. Unfortunately, one of my concerns was not addressed as I had intended, possibly due to a lack of detail on my part. I had suggested that wild-type hyphae be shown with calcofluor white staining side-by-side with the two kinase mutants to show what normal septa would look like and validate the conclusion that the mutants lack septa in their hyperpolarized structures. The authors added new Figure S9 to address this, but unfortunately it includes Cdc3-GFP signal demarking the septin rings, which overlaps with and occludes the calcofluor white signal at the septa. This figure also does not use the same imaging and image processing method as the figures I was referring to - Figure 5A and Figure 8C. I still think it is important that wild-type hyphal controls (just SC5314) be included specifically for these figures, highlighting normal septum signal with this imaging procedure.

**Have all data underlying the figures and results presented in the manuscript been provided?**

Large-scale datasets should be made available via a public repository as described in the *PLOS Genetics*    data availability policy, and numerical data that underlies graphs or summary statistics should be provided in spreadsheet form as supporting information., and numerical data that underlies graphs or summary statistics should be provided in spreadsheet form as supporting information., and numerical data that underlies graphs or summary statistics should be provided in spreadsheet form as supporting information., and numerical data that underlies graphs or summary statistics should be provided in spreadsheet form as supporting information.

Reviewer #1: Yes

Reviewer #3: Yes

PLOS authors have the option to publish the peer review history of their article (what does this mean?). If published, this will include your full peer review and any attached files.). If published, this will include your full peer review and any attached files.). If published, this will include your full peer review and any attached files.). If published, this will include your full peer review and any attached files.

**Do you want your identity to be public for this peer review?** For information about this choice, including consent withdrawal, please see our  For information about this choice, including consent withdrawal, please see our  For information about this choice, including consent withdrawal, please see our  For information about this choice, including consent withdrawal, please see our Privacy Policy....

Reviewer #1: No

Reviewer #3: No

**Figure resubmission:**
---

## [Editor Report · Decision Letter 2]

2 Apr 2026

Dear Dr Morschhäuser,

We are pleased to inform you that your manuscript entitled "Inducible gene deletion reveals essentiality of protein kinases and a septation initiation network in Candida albicans" has been editorially accepted for publication in PLOS Genetics. Congratulations!

Yours sincerely,

Jennifer A Benanti

Academic Editor

PLOS Genetics

Geraldine Butler

Section Editor

PLOS Genetics

Aimée Dudley

Editor-in-Chief

PLOS Genetics

Anne Goriely

Editor-in-Chief

PLOS Genetics

BlueSky: @plos.bsky.social

Comments from the reviewers (if applicable):

**Data Deposition**

If you have submitted a Research Article or Front Matter that has associated data that are not suitable for deposition in a subject-specific public repository (such as GenBank or ArrayExpress), one way to make that data available is to deposit it in the Dryad Digital Repository. As you may recall, we ask all authors to agree to make data available; this is one way to achieve that. A full list of recommended repositories can be found on our . As you may recall, we ask all authors to agree to make data available; this is one way to achieve that. A full list of recommended repositories can be found on our . As you may recall, we ask all authors to agree to make data available; this is one way to achieve that. A full list of recommended repositories can be found on our . As you may recall, we ask all authors to agree to make data available; this is one way to achieve that. A full list of recommended repositories can be found on our website....

http://datadryad.org/submit?journalID=pgenetics&manu=PGENETICS-D-25-01229R2

Additionally, please be aware that our data availability policy requires that all numerical data underlying display items are included with the submission, and you will need to provide this before we can formally accept your manuscript, if not already present. requires that all numerical data underlying display items are included with the submission, and you will need to provide this before we can formally accept your manuscript, if not already present. requires that all numerical data underlying display items are included with the submission, and you will need to provide this before we can formally accept your manuscript, if not already present. requires that all numerical data underlying display items are included with the submission, and you will need to provide this before we can formally accept your manuscript, if not already present.

**Press Queries**

If you or your institution will be preparing press materials for this manuscript, or if you need to know your paper's publication date for media purposes, please inform the journal staff as soon as possible so that your submission can be scheduled accordingly. Your manuscript will remain under a strict press embargo until the publication date and time. This means an early version of your manuscript will not be published ahead of your final version. PLOS Genetics may also choose to issue a press release for your article. If there's anything the journal should know or you'd like more information, please get in touch via plosgenetics@plos.org....

---

## [Editor Report · Acceptance letter]

PGENETICS-D-25-01229R2

Inducible gene deletion reveals essentiality of protein kinases and a septation initiation network in Candida albicans

Dear Dr Morschhäuser,

We are pleased to inform you that your manuscript entitled "Inducible gene deletion reveals essentiality of protein kinases and a septation initiation network in Candida albicans" has been formally accepted for publication in PLOS Genetics! Your manuscript is now with our production department and you will be notified of the publication date in due course.

With kind regards,

Anita Estes

PLOS Genetics

On behalf of:
